# Endangered predators and endangered prey: Seasonal diet of Southern Resident killer whales

M. Bradley Hanson[1]*, Candice K. Emmons[1], Michael J. Ford[1], Meredith Everett[1], Kim Parsons[1], Linda K. Park[1], Jennifer Hempelmann[1], Donald M. Van Doornik[2], Gregory S. Schorr[3], Jeffrey K. Jacobsen[4], Mark F. Sears[1], Maya S. Sears[1], John G. Sneva[5], Robin W. Baird[6], Lynne Barre[7]

1 Conservation Biology Division, Northwest Fisheries Science Center, National Marine Fisheries Service, National Oceanic and Atmospheric Administration, Seattle, Washington, United States of America, 2 Conservation Biology Division, Northwest Fisheries Science Center, National Marine Fisheries Service, National Oceanic and Atmospheric Administration, Manchester Research Station, Manchester, Washington, United States of America, 3 Marine Ecology and Telemetry Research, Seabeck, Washington, United States of America, 4 Bio-Waves, Incorporated, Encinitas, California, United States of America, 5 Washington Department of Fish and Wildlife, Olympia, Washington, United States of America, 6 Cascadia Research Collective, Olympia, Washington, United States of America, 7 Protected Resources Division, West Coast Regional Office, National Marine Fisheries Service, National Oceanic and Atmospheric Administration, Seattle, Washington, United States of America

* brad.hanson@noaa.gov

**Data Availability Statement:** Reference sequences developed for this paper are available at GenBank with accession numbers provided in S2 Table. DNA sequence files (fasta) obtained from the fecal

## Abstract

Understanding diet is critical for conservation of endangered predators. Southern Resident killer whales (SRKW) (*Orcinus orca*) are an endangered population occurring primarily along the outer coast and inland waters of Washington and British Columbia. Insufficient prey has been identified as a factor limiting their recovery, so a clear understanding of their seasonal diet is a high conservation priority. Previous studies have shown that their summer diet in inland waters consists primarily of Chinook salmon (*Oncorhynchus tshawytscha*), despite that species' rarity compared to some other salmonids. During other times of the year, when occurrence patterns include other portions of their range, their diet remains largely unknown. To address this data gap, we collected feces and prey remains from October to May 2004–2017 in both the Salish Sea and outer coast waters. Using visual and genetic species identification for prey remains and genetic approaches for fecal samples, we characterized the diet of the SRKWs in fall, winter, and spring. Chinook salmon were identified as an important prey item year-round, averaging ~50% of their diet in the fall, increasing to 70–80% in the mid-winter/early spring, and increasing to nearly 100% in the spring. Other salmon species and non-salmonid fishes, also made substantial dietary contributions. The relatively high species diversity in winter suggested a possible lack of Chinook salmon, probably due to seasonally lower densities, based on SRKW's proclivity to selectively consume this species in other seasons. A wide diversity of Chinook salmon stocks were consumed, many of which are also at risk. Although outer coast Chinook samples included 14 stocks, four rivers systems accounted for over 90% of samples, predominantly the Columbia River.

samples and control groups will be deposited in the Dryad data repository prior to publication. All other data are included in the paper or in the supplemental tables. https://doi.org/10.5061/dryad.sn02v6x35

**Funding:** Funding was provided by the Northwest Fisheries Science Center, National Marine Fisheries Service, National Oceanic and Atmospheric Administration, https://www.nwfsc.noaa.gov/; MBH received the following awards from United States Navy Pacific Fleet: N00070-14-MP-4C762, N00070-15-MP-4C363, N00070-16-MP-4C872, https://www.cpf.navy.mil/. The funders had no role in study design, data collection and analysis, decision to publish, or preparation of the manuscript. NOAA provided funding to Biowaves Inc. for part of this study and JKJ was subcontracted by them to conduct some data collection. Biowaves Inc. provided no funding for the study nor did it play a role in the study design, data analysis, decision to publish, or preparation of the manuscript and only provided financial support in the form of authors' salaries and/or research materials.

**Competing interests:** Our affiliation with Biowaves, Inc. does not alter our adherence to PLOS ONE policies on sharing data and materials.

Increasing the abundance of Chinook salmon stocks that inhabit the whales' winter range may be an effective conservation strategy for this population.

## Introduction

Understanding the seasonal diet of wildlife is important for both the conservation of predators and for understanding effects on their prey. Evaluating predator-prey dynamics is particularly important when one, or both, is an endangered or at-risk species. There are examples of at-risk predator populations consuming relatively robust prey populations (e.g., Steller sea lions, (*Eumatopias jubatus*) consuming cod and pollock [1]) and of robust predator populations eating threatened prey (e.g., pinnipeds consuming Pacific salmon [2] or Atlantic cod (*Gadus morhua*) [3, 4]). Management efforts to support adequate prey abundance for an at-risk predator becomes even more complex when their preferred prey are also commercially exploited, as is often the case for fish consumed by marine mammals.

Several adjacent communities of fish-eating killer whales (*Orcinus orca*) inhabit the outer coast and inland waters of the eastern North Pacific Ocean [5, 6]. The two groups that occur near the south-end of this distribution, the Northern and Southern Resident-type killer whales, have distinct but partially overlapping ranges that center on Vancouver Island (Fig 1, [7]). Both populations are thought to be salmon specialists, especially Chinook salmon (*Oncorhynchus tshawytscha*), for nearly all of their summer diet [8–11]. However, the population trajectories of these populations have been quite different since the initiation of photo-identification studies in the mid-1970s [7]. The Northern Resident-type killer whale (NRKW) population, numbering about 120 whales in the 1970s, increased nearly continuously at about 2%, now numbering about 300 whales [12]. Conversely, although the southern-most group of fish-eating killer whales, the Southern Resident killer whales (SRKW) did gradually increase though the mid-1990s, a 20% decline that occurred through 2001 prompted the listing of the population under the U.S. Endangered Species Act (ESA) and Canadian Species at risk Act (SARA) and the population as fluctuated since then. SRKWs provides an example of an endangered [13] predator whose diet largely comprises a threatened prey species. The SRKW population currently consists of 74 individual whales [14] and is comprised of three largely matrilineal groups, referred to as pods with alphabetic identifications (i.e., J, K and L; [15]). Most populations of Chinook salmon along the west coast of the United States are themselves listed as threatened or endangered under the ESA [16], and are harvested in commercial, recreational, and tribal fisheries. Simultaneously achieving recovery of this predator and its prey while sustaining viable fisheries presents a significant management challenge [2, 17–19].

Lack of sufficient prey availability is a significant risk factor for the SRKW population [20–22]. Specifically, low Chinook salmon abundance has been associated with low killer whale fecundity and survival [23–25]. Analysis of both prey remains [8, 9, 11] and fecal samples [10, 11] have provided complementary approaches for characterizing these killer whales' diet. However, to date, these studies have been limited to the summer months when the whales' range is generally confined to the Salish Sea [8–11] (Fig 1). This season and location coincides with the seasonal return of adult Chinook and coho salmon (*Oncorhynchus kisutch*) to their natal rivers to spawn, when the fish are found in relatively high densities in the narrow passages of the inland waters [26] (Fig 1).

The diet of the SRKWs during the fall/early winter (October-early January), mid-winter/early spring (late January-April), and spring (May) when both whales and salmon are likely

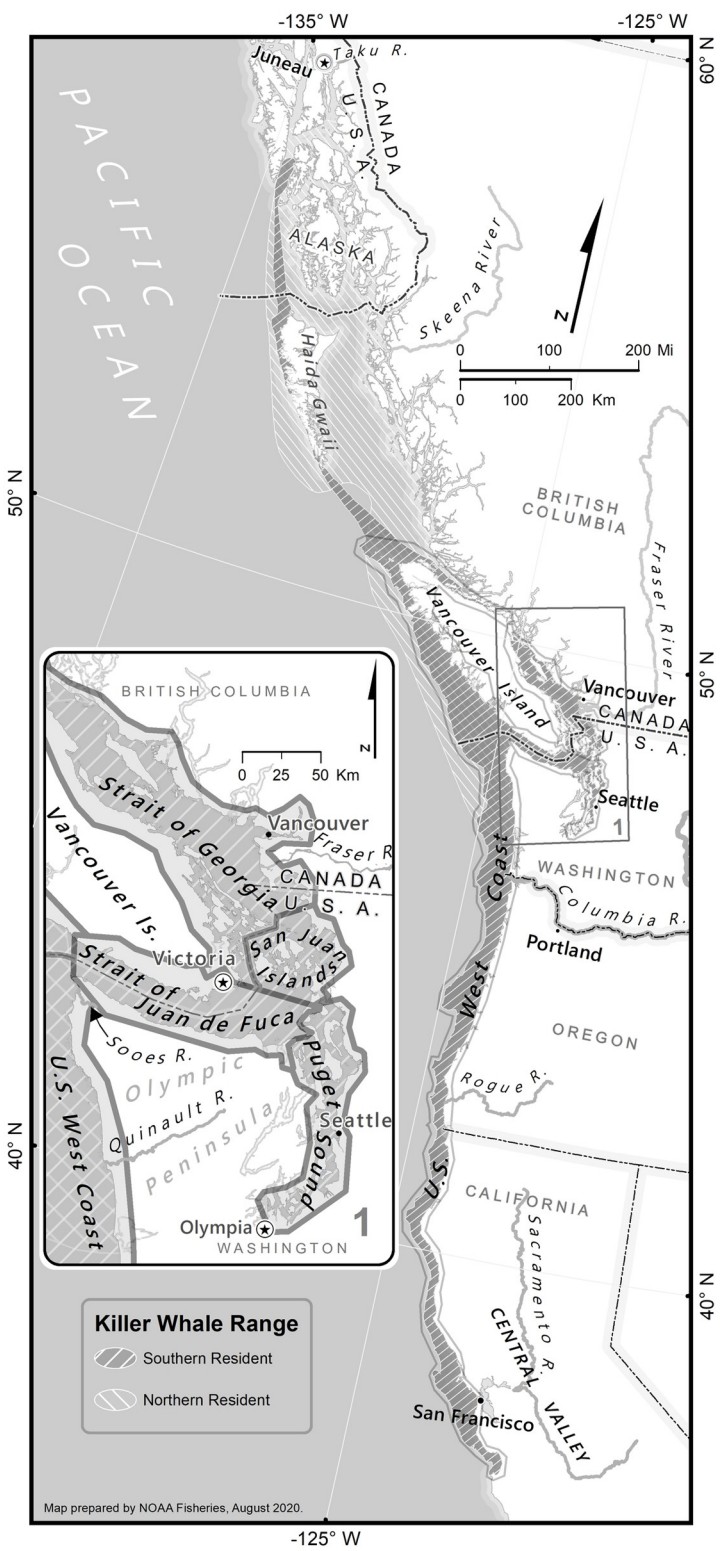

**Fig 1. Ranges of Resident-type killer whales in the eastern north Pacific Ocean and study area for prey and fecal samples collected from Southern Resident killer whales between October and May from 2004 to 2017.**

dispersed over a greater geographic area, is largely unknown and represents a critical data need for recovery of this population. A comprehensive understanding of seasonality in these whales' diet is important for evaluating prey preferences, needs, and availability, a risk factor outlined in this species' Recovery Plan [22], as well as gaining insights into the degree to which they switch to other prey and thus their resiliency to the potential for declining preferred prey. This information will be useful for appropriately targeting management efforts aimed at increasing the prey base for the SRKWs. Here we address this critical need by characterizing the SRKW's diet during fall, winter, and spring throughout their range (Fig 1), including the outer coast waters along the U.S. west coast and most regions of the Salish Sea. We further extended the analyses to genetically identify the stocks of origin for Chinook salmon prey. The results of this study will provide information that is of particular value to managers developing targeted management strategies for these prey species.

## Methods

### Field methods and sampling

We collected remains from prey captures (fish scales and/or bits of tissue) and fecal samples from SRKWs during both dedicated and opportunistic field efforts between October and May from 2004 through 2017, using techniques detailed in [10, 11]. Sample collection was conducted using small boats deployed daily from land or from a larger ship, depending on time of year and location. Whales present during sampling were photographed and identified to individual [27] and pod, allowing for the results to be presented for each pod separately because these pods occupy different ranges, and thus may consume different prey.

### Genetic analyses

Fish scales were air-dried and visually examined at 48X magnification to determine species and age (for salmon) from marine and freshwater annuli patterns [28, 29]. Genomic DNA was extracted from prey fish scales and tissue remains using standard methods (see [11]). Salmonid prey remains were identified visually (e.g., scale size and shape, tissue color), and species determined by PCR amplification and sequencing the COIII/ND3 region of the mitochondrial genome using primers and PCR reaction conditions described in [30]. Genetic stock identification methods were applied to determine stock identity of Chinook salmon prey remains samples. Specifically, sample genotypes at 13 nuclear microsatellite DNA loci were compared to a coast-wide data set of genotypes representing 42 Chinook salmon populations [31]. Prey remains samples were grouped into regional stock groups based upon genetic similarity [31], and the estimated contributions of these genetic stock groups to overall prey remains sample were estimated using the classification method of [32] as implemented in the program ONCOR [33]. The result was an estimate of the percentage (and standard error) of each stock group within the prey remains samples.

Prey species representation in fecal samples was based on targeted amplicon sequencing of an ~330bp region of the mitochondrial 16s gene on an Illumina MiSeq platform, as described previously [10]. Thirty fecal samples individually (one library per sample) and an additional 51 fecal samples were combined into 6 pooled libraries (2–17 fecal samples per library) due to available resources at the time these analyses were conducted (S1 Table). Pooled samples were normalized to contain an equimolar mix of mitochondrial prey DNA from each fecal sample using the qPCR method described in [10]. To evaluate the effects of pooling, one pool (comprising five fecal samples) was analyzed both pooled and individually. To assess variation due to DNA extraction and PCR variability eight samples were extracted and analyzed in duplicate, and four samples were analyzed in triplicate starting from the PCR step.

Three control samples were constructed using pre-determined proportions of vouchered target species mtDNA, each of which was replicated three times. Control 1 consisted of 25% each Pacific halibut (*Hippoglossus stenolepis*), lingcod (*Ophiodon elongatus*), Chinook salmon, and coho salmon (*O. kisutch*). Control 2 consisted of 40% each halibut and lingcod, 5% Pacific herring (*Clupea pallasii*), and 15% Chinook salmon. Control 3 consisted of 20% each Chinook and coho salmon, and 15% each steelhead (*O. mykiss*), sockeye (*O. nerka*), chum (*O. keta*), and pink salmon (*O. gorbuscha*). For targeted amplicon sequencing, each fecal sample, fecal pool and prey control were individually barcoded using Illumina indices prior to sequencing.

Illumina MiSeq reads were de-multiplexed, and trimmed to remove adaptor and index sequences prior to analysis, and forward and reverse reads were aligned and merged into single sequences using the PANDAseq software [34]. Length distributions of merged reads were analyzed using the Rsamtools package [35], and unusually long sequences (>400bp) were removed as likely PCR artifacts.

Merged reads were aligned against a custom reference sequence database using the BLAST + program [36]. Due to greater uncertainty in the whales' non-summer diet, we expanded the 19-species database used for the summer diet study to include additional fish species found on the U.S. west coast, for a total of 403 sequences from 246 species (S2 Table). All species and sequence data included in the reference baseline were identified and vouchered as part of the joint NWFSC/University of Washington fish voucher collection [37]. Diet composition was estimated separately for each library by counting the number of sequences assigned to each species in the reference database, after removing any host (killer whale) sequences.

Due to the non-random and opportunistic nature of sampling, we focused on describing diet patterns rather than statistical hypothesis testing. We summarized diet by region (Puget Sound, Juan de Fuca Strait and San Juan Islands, northern Georgia Strait, and outer coast waters of the U.S. west coast, Fig 1) and season (fall/early winter: October- early January; mid-winter/early spring: late January-April, and spring: May). Most of the "seasons" we defined correspond with some component of seasonal SRKW's range occupancy patterns, i.e., fall/early winter–Puget Sound, Juan de Fuca Strait and San Juan Islands; mid-winter /early spring–northern Georgia Strait, U.S. west coast waters. Species diversity from fecal samples was simply described as the number of species present in a sample, and as Simpson diversity, the probability of drawing two different species from a sample, calculated as $1 - \sum x_i^2$, where $x_i$ is the proportion of species $i$ in the sample. For both measures, only species present at >1% reads in at least one fecal sample were included in the analysis.

Each fecal sample was genotyped at either a series of microsatellite loci (as described in [38]) or single nucleotide polymorphism (SNP) loci (as in [39]) to genetically identify the whale from which it originated and enable analyses by pod. Work was conducted in U.S. waters under NMFS General Authorization No. 781–1725, and NMFS Scientific Research Permits 781–1824, and 16163. Work was conducted in Canadian waters under Marine Mammal License numbers MML 2006-02/SARA-24, MML 2007-03/SARA-64, MML 2008-03/SARA-84, MML 2012–03 SARA-84, License Number: XMMS 8 2014, File Number: 2014–22 SARA-355, License Number: XMMS 8 2014—Amendment 1, File Number: 2014–22 SARA-355.

Sample collection methods were approved by the NWFSC/AFSC Institutional Animal Care and Use Committee under protocol s A/NW2014-1, and A/NW 2015–2.

## Results

Southern Resident killer whales were encountered on 156 days between October and May from 2004 to 2017 in three areas of the Salish Sea (Puget Sound: 108 days, Juan de Fuca Strait/San Juan Islands (JdF/SJI, 9 days), and northern Georgia Strait (NGS, 3 days), and in outer

coast waters of Washington, Oregon, and California (36 days) (S3 Table and Fig 1). During these effort days we were able to detect and collect 81 fecal samples and observed 152 distinct prey capture events that yielded scales or tissue. These prey capture events yielded a total of 155 unique fish, i.e., some of the prey capture events each yielded two unique fish and some sample identifications failed (S1 and S3 Tables and Fig 2).

Most prey (64.5%) and fecal (65.4%) samples, were collected in the Salish Sea, and predominantly in the main basin of Puget Sound (61.3% prey, 59.3%, fecal, S3 Table). The majority of the remaining samples (35.3% prey, 34.6% fecal) were collected in outer coast waters, particularly off the Washington coast (S3 Table).

Prey remains were collected from all three pods in Puget Sound, although nearly twice as many samples were collected from J (19/20.4%) and K (24/25.8%) pods than L pod (11/11.8%). Thirty-nine prey remains samples (41.9%) were collected when some combination of J, K, and or L pods were together. Fecal samples collected in Puget Sound represented only J and K pods, but were mostly (83.7%) J pod (S1 Table). Prey remains samples collected in the JdF/SJI area were from only J (1/33.3%) and L (2/66.7%) pods but fecal samples from this area included members all three pods. Both prey remains samples collected in the northern Georgia Strait were from J pod. The outer coast prey remains samples were mostly collected when both K and L where together (40/74.1%) although a few (4/7.4%) where from either J or L pods. Outer coast fecal samples were only collected from K and L pods (S1 Table).

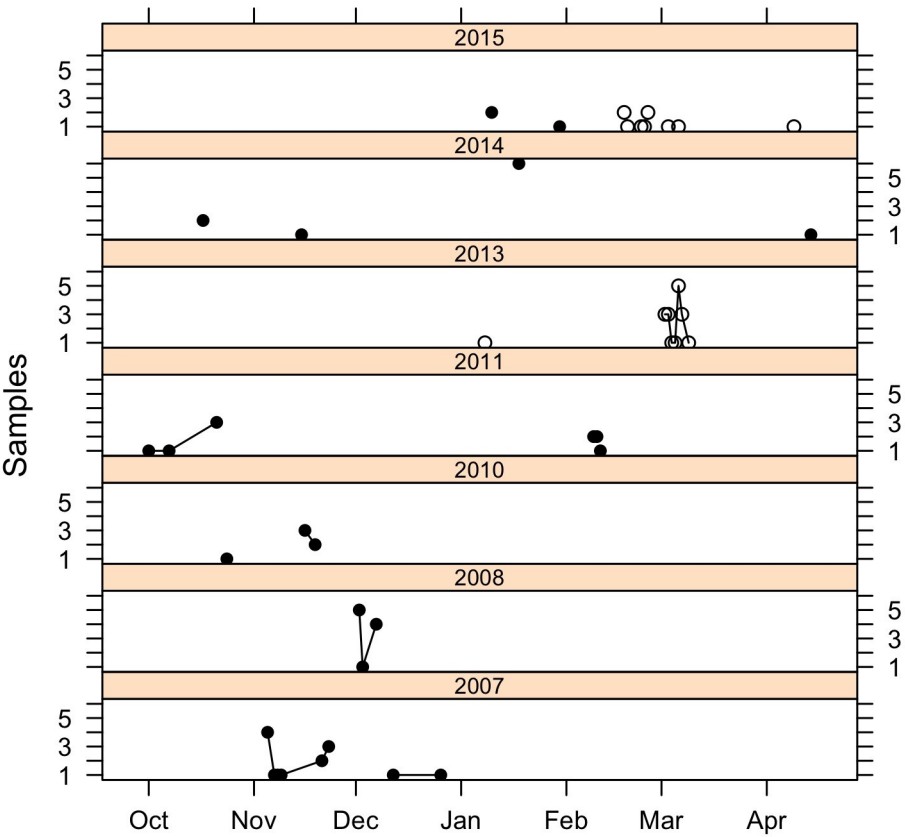

**Fig 2. Months and years of Southern Resident killer whale fecal sample collections.** Y-axis indicates number of samples collected per day that were included in the analysis. Open and filled symbols indicate outer coast and Salish Sea samples, respectively. Fecal samples that were pooled prior to sequencing are connected by lines.

## Species composition diet from fecal samples

After aligning and merging forward and reverse reads from the 16S mtDNA amplicon sequencing, we obtained a total of 15,874,848 sequences, of which 15,468,815 (97.4%) were <400bp and used for species identification. Among the 63 libraries (including controls and replicates), the number of reads ranged from 39,337 to 814,515 and averaged 245,537 per library. The proportion of host (killer whale) sequence among samples averaged 12.5% (range 0–92%). After removal of host sequences, the number of sequences used for diet analysis averaged 219,500 per library and ranged from 19,170 to 814,200. Nearly all sequences (99.8%) aligned to one or more sequences in the reference database, with a mean sequence identity of 99.4%. Identical blast scores between two or more species were very rare (<0.1%) and therefore were ignored in downstream analyses.

The observed mixtures in the known composition controls were generally close to expectations (S4 Table). For controls 1 and 2, the mean estimates differed from the expected values by a maximum of 4 percentage points, with halibut in control 1 at a somewhat higher proportion than expected (29% versus 25%). In control 2, the Chinook salmon proportion was slightly lower than expected (11% versus 15%). In control 3, differences between observed and expected proportions of up to 6 percentage points were observed, with steelhead more abundant than expected (21% versus 15%) and sockeye salmon less abundant than expected (9% versus 15%). For all three controls, there was little variation in estimates among replicates, with standard errors all < 0.005 (S4 Table).

Similar to the controls, the results among replicate samples varied little, with all standard errors < 0.05 and most < 0.01, indicating that random variation associated with DNA extraction, PCR, and sequencing was not a major source of variation among samples (S1 Table). Neither the number nor diversity of species per sample differed significantly between the pooled and single fecal samples (Welch two sample t-test; p = 0.76 and 0.45, respectively), so these groups were not distinguished in further analyses.

Chinook salmon was the most common prey species when averaged across all fecal samples in each of two areas were most samples were collected (51.0%, 67.3%), Puget Sound and outer coast waters, respectively (S1 Table). Chum salmon was the next most common species consumed in two areas of the three areas (Puget Sound, 31.2%, JdF/SJI 31.5%) but virtually nonexistent in outer coast waters (1.2%) (S1 Table). Although coho was predominant in the JdF/SJI samples (53.8%) it was a very minor contributor in Puget Sound (0.7%) and outer coast waters (0.1%) (S1 Table). Steelhead occurred primarily in the outer coast samples (8.7%) and Puget Sound (3.5%) (S1 Table). Non-salmonids made up 22.7%, 12.5% and 10.6% in the outer coast waters, Puget Sound and JdF/SJI samples, respectively (S1 Table). Of the non-salmonid samples, Lingcod was consistently prominent in two areas, outer coast waters (14.8%), and Puget Sound (5.2%) followed by halibut (7.3%, outer coast waters) and big skate (*Rana binoculata*, 4.3%, Puget Sound) (S1 Table).

Prey species composition varied considerably among fecal samples. The number of prey species represented by > 1% of prey sequences ranged from 1 to 5 different fish species. Diet diversity measured as either species number or Simpson diversity was highest in winter (Fig 3). Chinook, chum salmon, and big skate were each present at >95% in at least one fecal sample, and several other species (coho salmon, steelhead, English sole (*Parophrys vetulus*), halibut, and lingcod) were present at >30% in at least one sample (S1 Table). Seasonally, Chinook salmon were present in most samples, with all outer coast water's samples containing some proportion (14–99%) of prey sequences generated from fecal sample analysis, and most samples in Puget Sound containing this species (0–100%) (S1 Table and Fig 4). Chum salmon were mostly prevalent from samples in the fall/early winter in Puget Sound (0–99%), but were largely absent from

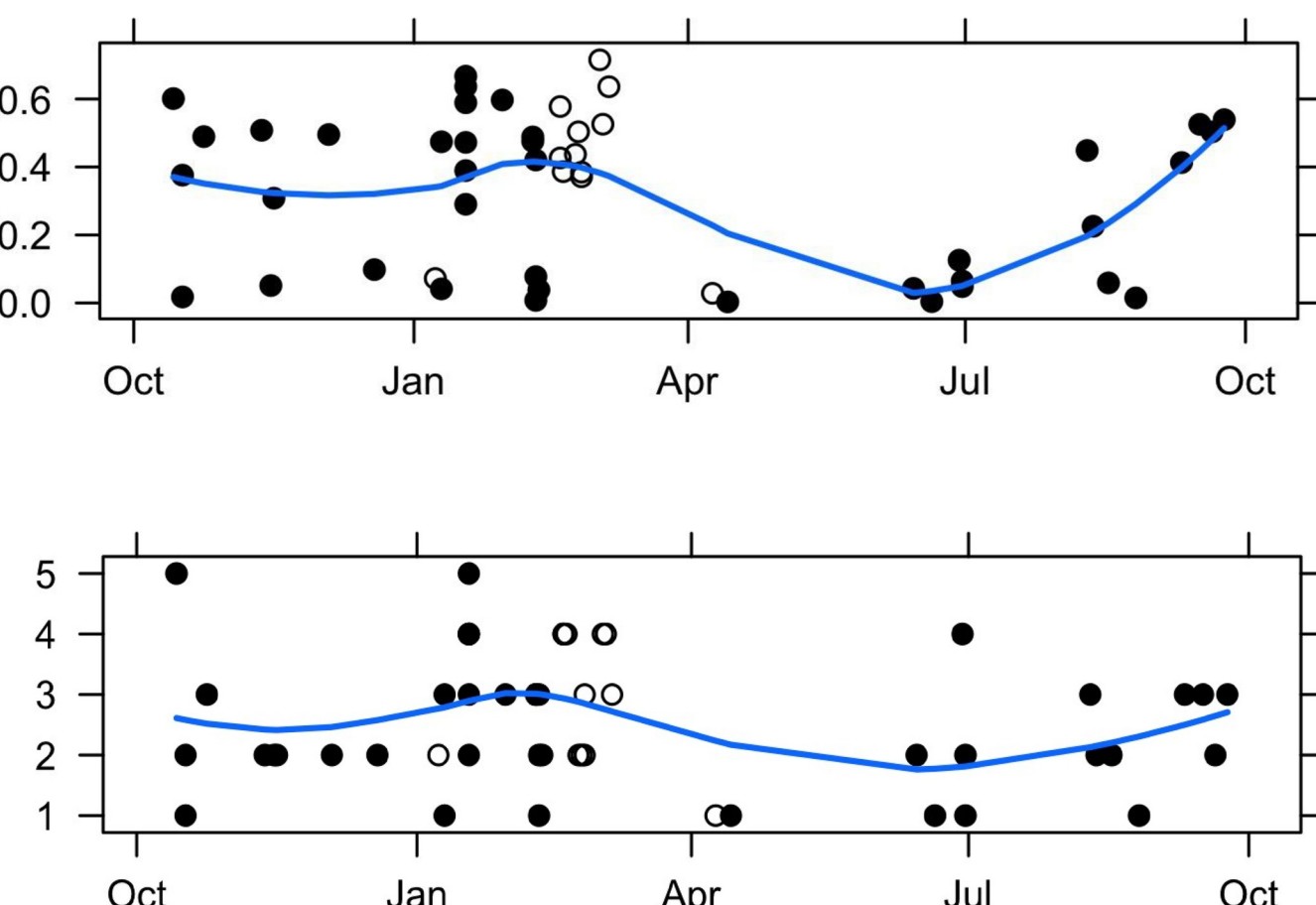

**Fig 3. Seasonal trends in the diversity (top) and number (bottom) of Southern Resident killer whale prey species (>1%) per fecal sample.** Open and filled symbols indicate samples from the outer coast and Salish Sea, respectively. The line indicates the predicted values from local polynomial regression. Data from May—September are from Ford et al. (2016).

the samples collected in winter and spring (only one sample, 12%) in outer coast waters (S1 Table and Fig 4). Big skate was present at >95% in one sample in January, but were absent or very low (<1%) in all other samples (S1 Table). Several species, including steelhead, lingcod, and several flatfish (e.g., Dover sole (*Microstomus pacificus*), arrowtooth flounder (*Atheresthes stomias*)) were present in appreciable frequencies only in winter months (January to early March) in outer coast waters (S1 Table and Fig 4). Steelhead was present in quantities up to 34% in five of 12 fecalsamples from February and March in outer coast waters. Halibut was as high as 32% in eight of the 12 February/March fecal samples, and lingcod was found in nine fecal samples as high as 42%, all in outer coast waters (S1 Table). Coho salmon was only found in October and was the major component (53.8%) of part of a set of 5 pooled fecal samples from the JdF/SJI area but only represented 5.6% of sequences over all fecal samples (S1 Table).

## Prey composition from predation event samples

The 152 prey remains samples collected across the four regions yielded 155 results (S3 Table). Most were salmonids, the majority being Chinook (72) and chum (54) salmon (Table 1). Most of the samples (93) were collected in Puget Sound, from which 95 fish could be identified to species, with five samples indicating the presence of two fish (Tables 1 and S3). The majority of salmon in

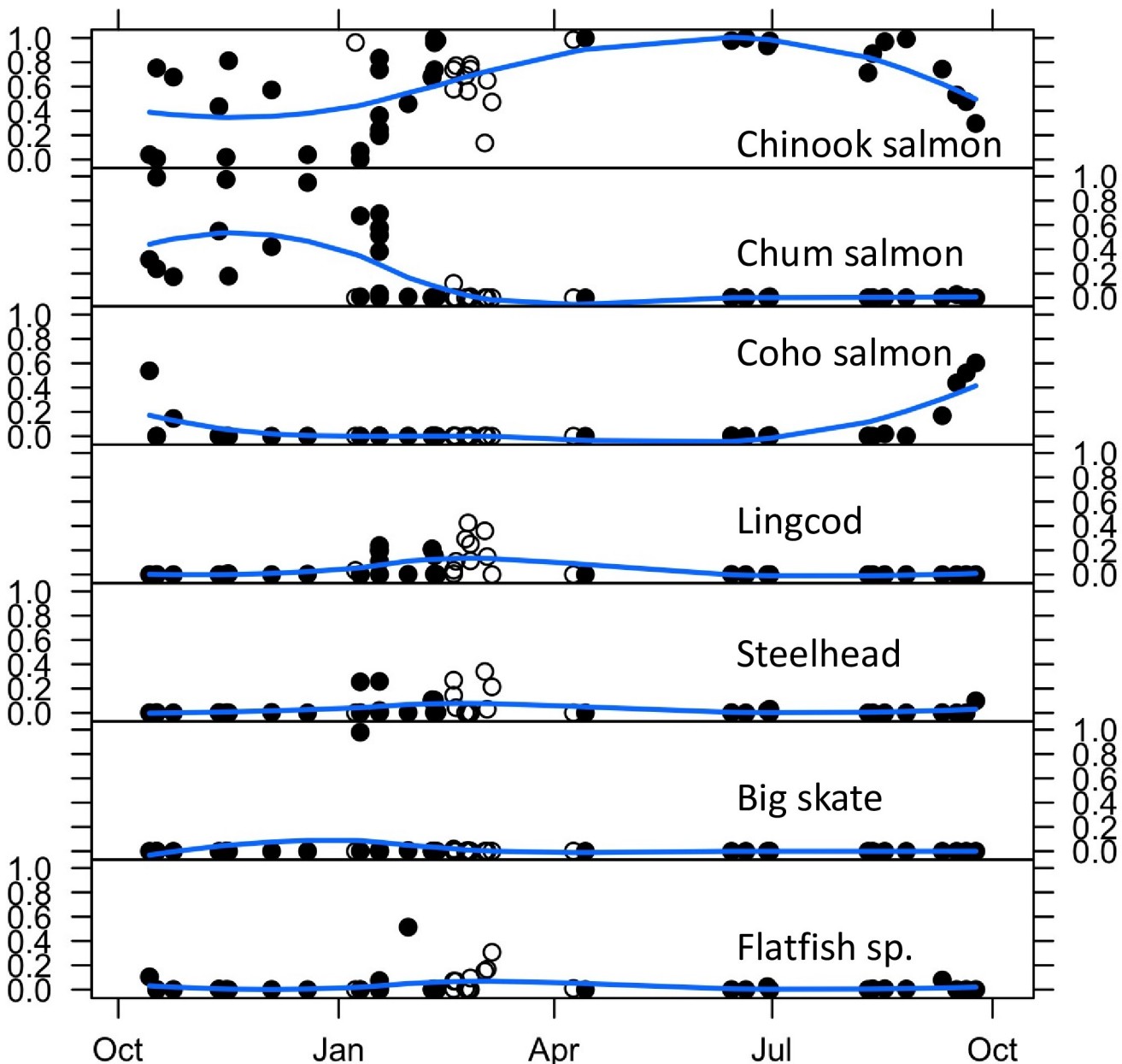

**Fig 4. Seasonal proportion of sequences assigned to seven Southern Resident killer whale prey species (or groups of species) across all fecal samples.**
Open and filled symbols indicate samples from the outer coast and Salish Sea, respectively. Lines indicate predicted values from local polynomial regression. Data from May—September are from [10].

these samples in the fall/early winter in Puget Sound were chum salmon (60.2%, n = 53), with smaller proportions of Chinook (22.7% n = 20), coho salmon (12.5%, n = 11), and steelhead (4.5%, n = 4) present (Table 1). All six prey remains samples collected in the spring were collected in Puget Sound, yielding seven results, all Chinook salmon. Both of the prey remains samples collected in the northern Strait of Georgia in mid-winter/early spring were Chinook salmon (Table 1,). The three prey remains samples collected in the JdF/SJI area in October and April were identified as coho salmon, steelhead, and halibut. In outer coast waters, we collected 54 prey remains samples yielding 55 prey items (S3 and 1 Tables). Three prey remains samples were

**Table 1. Total number of Southern Resident killer whale prey identified to species from Scale (S) or Tissue (T), and Fecal (F) analyses (excluding species with < 1% in weighted averages) by season in Puget Sound (PS), Juan de Fuca Strait/San Juan Islands (JdF/SJI), Northern Georgia Strait (NGS), and outer Coast Waters (CW).**

| Prey Species | Location | | Season | | | Total |
|---|---|---|---|---|---|---|
| | | | Fall/early winter | Mid -winter/early spring | Spring | |
| Chinook | PS | S/T | 19 | 1 | 7 | 27 |
| | PS | F | ≥15 | 6 | 0 | ≥21 |
| | NGS | S/T | | 2 | | 2 |
| | NGS | F | | 0 | | 0 |
| | JdF/SJI | S/T | 0 | | | 0 |
| | JdF/SJI | F | ≥1 | | | ≥1 |
| | CW | S/T | | 43 | | 43 |
| | CW | F | | ≥13 | | ≥13 |
| Chum | PS | S/T | 53 | 0 | | 53 |
| | PS | F | ≥15 | 0 | | ≥15 |
| | NGS | S/T | | 0 | | 0 |
| | NGS | F | | 0 | | 0 |
| | JdF/SJI | S/T | 0 | | | 0 |
| | JdF/SJI | F | ≥1 | | | ≥1 |
| | CW | S/T | | 1 | | 1 |
| | CW | F | | 1 | | 1 |
| Coho | PS | S/T | 10 | 1 | | 11 |
| | PS | F | 1 | 0 | | 1 |
| | NGS | S/T | | 0 | | 0 |
| | NGS | F | | 0 | | 0 |
| | JdF/SJI | S/T | 1 | | | 1 |
| | JdF/SJI | F | ≥1 | | | ≥1 |
| | CW | S/T | | 0 | | 0 |
| | CW | F | | 0 | | 0 |
| Steelhead | PS | S/T | 3 | 1 | | 3 |
| | PS | F | 3 | 2 | | 5 |
| | NGS | S/T | | 0 | | 0 |
| | NGS | F | | 0 | | 0 |
| | SJI/JdF | S/T | 1 | | | 1 |
| | SJI/JdF | F | 0 | | | 0 |
| | CW | S/T | | 10 | | 10 |
| | CW | F | | ≥6 | | ≥6 |
| Halibut | PS | S/T | 0 | 0 | | 0 |
| | PS | F | 0 | 0 | | 0 |
| | NGS | S/T | | 0 | | 0 |
| | NGS | F | | 0 | | 0 |
| | JdF/SJI | S/T | 1 | | | 1 |
| | JdF/SJI | F | ≥1 | | | ≥1 |
| | CW | S/T | | 1 | | 1 |
| | CW | F | | ≥7 | | ≥7 |
| Lingcod | PS | S/T | 0 | 0 | | 0 |
| | PS | F | 4 | 2 | | 6 |
| | NGS | S/T | | 0 | | 0 |
| | NGS | F | | 0 | | 0 |
| | JdF/SJI | S/T | 0 | | | 0 |

*(Continued)*

**Table 1.** (Continued)

| Prey Species | Location | | Season | | | Total |
|---|---|---|---|---|---|---|
| | | | Fall/early winter | Mid -winter/early spring | Spring | |
| | JdF/SJI | F | 0 | | | 0 |
| | CW | S/T | | 0 | | 0 |
| | CW | F | | ≥9 | | ≥9 |
| Big Skate | PS | S/T | 0 | 0 | | 0 |
| | PS | F | 1 | 0 | | 1 |
| | NGS | S/T | | 0 | | 0 |
| | NGS | F | | 0 | | 0 |
| | JdF/SJI | S/T | 0 | | | 0 |
| | JdF/SJI | F | 0 | | | 0 |
| | CW | S/T | | 0 | | 0 |
| | CW | F | | 0 | | 0 |
| Arrowtooth Flounder | PS | S/T | 0 | 0 | | 0 |
| | PS | F | 0 | 0 | | 0 |
| | NGS | S/T | | 0 | | 0 |
| | NGS | F | | 0 | | 0 |
| | JdF/SJI | S/T | 0 | | | 0 |
| | JdF/SJI | F | ≥1 | | | ≥1 |
| | CW | S/T | | 0 | | 0 |
| | CW | F | | 0 | | 0 |
| English sole | PS | S/T | 0 | 0 | | 0 |
| | PS | F | 3 | 0 | | 3 |
| | NGS | S/T | | 0 | | 0 |
| | NGS | F | | 0 | | 0 |
| | JdF/SJI | S/T | 0 | | | 0 |
| | JdF/SJI | F | 0 | | | 0 |
| | CW | S/T | | 0 | | 0 |
| | CW | F | | 0 | | 0 |
| Total S/T | | | 88 | 60 | 7 | 155 |
| Total F | | | ≥47 | ≥46 | 0 | ≥93 |

≥ indicates that species may have been present in one or more of the samples that were pooled, see S1 Table.

Blanks indicate no sampling effort.

collected in Northern California (all Chinook) and the balance were collected between northern Oregon and northern Washington. Forty-three (78.2%) were Chinook salmon, 10 (18.2%) were steelhead and one each was a chum salmon and a halibut (Table 1). Of the five samples collected off the northern Oregon coast, four were Chinook salmon and one was a steelhead. Twenty samples were collected off southwest Washington, between the mouth of the Columbia and the Quinault River, of which 19 were Chinook salmon and one was a steelhead. The 25 prey remains samples collected off the northern Washington coast between the Hoh and Soos Rivers consisted of 16 Chinook salmon, seven steelhead, one chum salmon, and one halibut.

## Genetic stock origin of Chinook salmon prey

The genetic stock of Chinook salmon from predation events could be determined for 20 of the 27 samples collected in Puget Sound (Table 2). Of these, 67.2% were estimated to have

**Table 2. Estimated mean percentage (±SE) of Chinook salmon stock composition of Southern Resident killer whale prey from Puget Sound and outer coast waters based on remains from prey capture event samples for fall/ early winter and Mid-winter/early spring.**

| | Area | |
| --- | --- | --- |
| | Puget Sound | Outer coast waters |
| | Season | |
| Genetic stock group | Fall/early winter | Mid-winter/early spring |
| N | 20 | 33 |
| Taku R. | | 3.4±2.8 |
| Upper Stikine R. | 0.0±2.6 | 1.0±2.9 |
| So. SEAk—Stikine | 0.0±7.0 | 0.0±1.3 |
| So. SEAk | 0.0±7.3 | 0.0±4.4 |
| Nass R. | 0.0±7.3 | |
| Upper Skeena R. | 4.4±4.0 | |
| Lower Skeena R. | 0.0±2.5 | |
| Cent. BC Coast | 0.0±7.3 | 0.0±1.2 |
| West Vancouver I. | | 0.0±2.2 |
| East Vancouver I. | 0.0±5.8 | 0.0±1.7 |
| Fraser R. (all) | 14.7 | 6.5 |
| Upper Fraser R. | | 0.0±1.9 |
| Mid Fraser R. | | 4.3±2.3 |
| No. Thompson R. | | |
| So. Thompson | | 0.0±1.8 |
| Lower Thompson R. | 4.7±4.9 | |
| Lower Fraser R. | 10.0±6.9 | 2.3±3.0 |
| Puget Sound (all) | 67.2 | 14.2 |
| No. Puget Sound | 5.3±11.1 | 4.9±5.6 |
| So. Puget Sound | 61.9±7.7 | 9.3±4.4 |
| Juan de Fuca St. | 0.0±6.8 | |
| Washington coast | 4.9±4.8 | |
| Columbia R. (all) | 1.9 | 53.6 |
| Snake R. (spring/summer) | | 2.2±3.4 |
| Snake R. (fall) | | 0.0±2.4 |
| Upper Col. R. (sum./fall) | 0.0±4.8 | 9.4±4.4 |
| Mid/Upper Col R. spring | | 4.6±3.4 |
| Mid. Col. R. Tule | 1.9±6.2 | 10.0±4.7 |
| Deschutes R. | | 0.0±4.3 |
| Willamette R. | | 0.0±1.5 |
| Lower Col. R. spring | | 17.5±6.6 |
| Lower Col. R. Fall | | 9.9±7.3 |
| No. Oregon Coast | 0.0±4.8 | |
| Mid. Oregon Coast | 0.0±10.0 | 0.0±4.6 |
| Rogue R. | 2.2±6.2 | 0.0±0.2 |
| Klamath R. | | 2.2±2.3 |
| Central Valley Total | 4.8 | 19.0 |
| Central Valley Spring | | 11.0±6.6 |
| Central Valley Fall | 4.8±4.9 | 8.0±8.3 |

Chinook genetic stock groups are arranged from north to south. Chinook populations from [26], Appendix 1.

originated from the two Puget Sound stocks (61.9%, South Puget Sound; 5.3% from North Puget Sound), and 14.7% from the Fraser River. The remaining 18.1% were assigned to genetic stocks well outside Puget Sound, including the upper Skeena River (4.4%), Washington outer coast rivers (4.9%), mid-Columbia Tule stocks (1.9%), Rogue River (2.2%) and California Central Valley fall-run (4.8%). Of the two Chinook salmon prey collected in northern Georgia Strait, one originated from the Taku River and the other from the lower Fraser River.

Genetic stock origin could be determined for 33 of the 44 Chinook salmon prey remains samples collected in outer coast waters of Washington, Oregon, and California (Table 2). The vast majority of these (93.3%) originated in four regions: the Columbia River (53.6%), the Central Valley (19.0%), Puget Sound (14.2%) and the Fraser River (6.5%). Of those originating in the Columbia River, about a third were from spring runs (adults returning to freshwater in spring) and two thirds were summer or fall-runs (adult return to freshwater in the summer or fall) (Table 2, see [31]). Chinook salmon consumed across winter months in outer coast waters tended to be mainly from fall- and summer-run stocks in the early part of winter, and spring-run stocks later in winter (S5 Table). Samples from six genetic stock groups were collected in February, most from the Columbia River, and in particular Lower Columbia fall-run stocks, and Upper Columbia summer- and fall-run stocks. In March, seven Columbia River stocks were consumed; spring-run Chinook salmon accounted for 30% of the stocks (S5 Table). This was also the earliest that Puget Sound Chinook salmon were observed in the outer coast samples. By April, Columbia River spring-run Chinook salmon were most prominent, mostly from the Middle/Upper Columbia River stock grouping (S5 Table).

Fish age based on scale annuli could be determined for 49 chum, 50 Chinook, and eight coho salmon as well as 11 steelhead (Table 3). Almost all of the chum salmon were sampled in Puget Sound, the majority of which (77.6%) were 4 years-old. Chinook salmon from Puget Sound were nearly equally distributed among ages 2, 3, and 4 years-old (Table 3). In outer coast waters, the majority of Chinook salmon (60.0%) were 4 years-old, with nearly twice as many 5 as 3 years-old (Table 3). Coho salmon were all 3 years of age and all the steelhead were 5 or 6 years old (Table 3).

## Discussion

### Species composition of diet

Our results, from both sample types, indicate that salmon, particularly Chinook salmon, represent a major component of the SRKW diet across all seasons and throughout a substantial portion of their range. Previous studies [8–11] found that Chinook salmon made up most of the whales' summer diet, and our results confirm that this species is a preferred, important, item throughout the year. However, results from fecal samples show a broader diet, particularly in winter, than results obtained from only scales and tissue. The tendency for selection of relatively rare, older age classes of Chinook salmon previously observed for resident-type killer whales [8], and our observation of a broader winter diet including non-salmonids likely available to these whales year-round, suggests a lack of their preferred prey during winter. While the average proportion of Chinook salmon in fall or winter fecal samples from this study was somewhat lower (49.1% in the Salish Sea, 67.3% on the outer coast, respectively) than observed in the summer (84%, [8]; 83%, [11]; 79%, [10]), our estimate of 80% Chinook salmon in the diet in outer coast waters based on prey remains in the winter and spring is nearly identical to that obtained for summer.

Our results also show that several other salmon species were present in the whales' fall/early winter diet, consistent with prior data from late summer [10]. Chum salmon make the next highest contribution to the diet (32.1% feces, 60.2% prey remains), specifically from October

**Table 3. Ages of salmonids and steelhead determined from scales of fish consumed by Southern Resident killer whales in different areas of their range.**

| Area | Species | Age class | | | | | | | | | | | | | | | | | |
|---|---|---|---|---|---|---|---|---|---|---|---|---|---|---|---|---|---|---|---|
| Puget Sound | | n | 0.1 | 0.2 | 1.1 | 0.3 | 1.2 | 2.1 | 0.4 | 1.3 | 2.2+ | R.2 | R.2+ | 0.5 | 1.4 | 2.3 | 3.2 | R.3 | 1.5 |
| | Age (years) | | 2 | 3 | | 4 | | | 5 | | | 5 or 6 | | 6 | | | | | 7 |
| | Chinook | 19 | 5 | 6 | 2 | 6 | | | | | | | | | | | | | |
| | Chum | 48 | | 4 | | 38 | | | | 7 | | | | | | | | | |
| | Coho | 7 | | | 7 | | | | | | | | | | | | | | |
| | Steelhead | 4 | | | | | | | | 1 | 1 | 1 | 1* | | | | | | |
| JDF/SJI | | | | | | | | | | | | | | | | | | | |
| | Chinook | | | | | | | | | | | | | | | | | | |
| | Chum | | | | | | | | | | | | | | | | | | |
| | Coho | 1 | | | 1 | | | | | | | | | | | | | | |
| | Steelhead | 1 | | | | | | | | 1 | | | | | | | | | |
| Northern Georgia Strait | | | | | | | | | | | | | | | | | | | |
| | Chinook | 2 | | | | 1 | | | | 1 | | | | | | | | | |
| | Chum | | | | | | | | | | | | | | | | | | |
| | Coho | | | | | | | | | | | | | | | | | | |
| | Steelhead | | | | | | | | | | | | | | | | | | |
| Outer coast | | | | | | | | | | | | | | | | | | | |
| | Chinook | 30 | | 4 | | 13 | 4 | | 3 | 4 | | | | | | | | | |
| | Chum | 1 | | | | 1 | | | | | | | | | | | | | |
| | Coho | | | | | | | | | | | | | | | | | | |
| | Steelhead | 6 | | | | | | | | | | 1* | | | | 2 | 1 | 2* | |

Age class given in European system, whereby years in freshwater after hatching and years in salt water are identified and separated by decimal point. Ages used elsewhere in this paper were obtained by summing the two European age values and adding 1 (e.g. 1.2 age converts to a 4th year fish). An "R" indicates that the freshwater age could not be estimated due to scale regeneration. Specific names of prey in Table 2.

* The total age of these steelhead is assumed to have a freshwater age of 2.

to January when the whales foraged in Puget Sound and JdF/SJI (31.5% feces), but only represents a trace in outer coast waters (1.2% feces, 1.8% prey remains). Our observation of a relatively high prevalence of chum salmon in SRKW diet in inland waters is consistent with previous observations of NRKWs in their northern Vancouver Island range [8]. It is perhaps not surprising that chum salmon comprised a relatively large proportion of the whales' diet when they were in Puget Sound as large runs of chum salmon return there in the fall due to extensive hatchery propagation [40]. Coho salmon were the next highest contributor in the fall/early winter in JdF/SJI (53.8%, feces) and Puget Sound (0.7% feces, 12.5% prey remains), which was consistent with previous observations of coho salmon in the whales' diet in late summer [10]. Chum and coho salmon have previously been reported in the diet of SRKW [8] as well as other North Pacific killer whale populations in British Columbia [8], Alaska [41], and Russia [42]. Steelhead were present in the killer whales' diet across seasons and areas, more prominently on the outer coast (8.7% feces, 18.2% prey remains) than in Puget Sound (3.5% feces and 4.5% prey remains). Previous studies in other seasons [8, 11, 43] and areas [8] also concluded that steelhead were potentially an important prey item for SRKWs. The nine steelhead that K and L pods captured in outer coast waters suggests they may be adept at taking advantage of this species, despite its relatively short residency time in close proximity of their natal rivers, associated with their migration to and from northern waters of the central North Pacific Ocean [44]. None of the more northern killer whale populations in British Columbia [8] or Alaska [41] have been documented to feed on steelhead.

Lingcod was the most abundant non-salmonid fish species identified in fecal samples, particularly in outer coast waters (14.8%, versus 5.2% in Puget Sound). This prey species was found in stomach contents of a killer whale carcass collected in the San Juan Islands (likely a resident type as salmon but no marine mammal parts were present, [45]). Lingcod was also documented in the stomach of a stranded killer whale in early winter, most likely from the northern resident community due its location in the Johnstone Strait area of British Columbia [9, 46]). Consequently, it was not unexpected that lingcod occurred in our outer coast fecal samples, particularly given their relatively high density in nearshore outer coast waters that are approximately 50-100m deep [47]. In addition, this species' life history may make male lingcod potentially more vulnerable to predation in winter than at other times of the year. Male lingcod make a directed spawning migration in the late fall and winter toward shore where they become territorial near rocky reef areas suitable for spawning [48, 49]. Their nests are reported to occur in waters <36m [50–52], which they guard for about 7 weeks [48], until about mid-April [50]. Halibut was the only non-salmonid collected from prey remains (one sample) and in feces that appeared more commonly on the coast (7.3%) than JdF/SJI (1.3%) or Puget Sound (0.1%). This species has been documented in the diet of resident killer whales in British Columbia [8, 9] and killer whales of unknown ecotype in Alaska [53]. Halibut is relatively abundant on the continental shelf of Washington [54], although a substantial portion of the population may migrate off the shelf during the winter [55]. Skate, found in high proportion in a single fecal sample, has not previously been identified as a prey item in SRKWs, although it has been reported in the diet of killer whales in Russia [42].

When combined with previous summer diet results [8–11], the picture that emerges is that although this population's diet contains Chinook salmon year-round, there are some distinct seasonal shifts (Figs 3 and 4). From June to August Chinook salmon is the nearly exclusive prey item [8–11] when SRKW are foraging in inland waters. By September their diet transitions to include up to 50% coho salmon when the whales are still in inland waters [10]. From October through December SRKW diet is comprised of a mix of coho, Chinook, and chum salmon (this study). During the period from January to March further diversity in diet occurs (at least when the whales are in outer coast waters) to include steelhead and various non-salmonid fishes (this study). By April and May the whales return to a mostly Chinook salmon diet (this study). There is also a general pattern of a higher proportion of non-salmonids in the diet along the coast compared to the inland waters (Fig 4). Although coho and chum salmon were consumed in inland waters in the fall, the near lack of these species in the winter diet of K/L pods in outer coast waters was most likely due to none of those species occurring in this area during this time of the year based on the migration patterns inherent to these species and stocks [56].

## Differences in diet results between prey remains and feces

We found a three-fold larger number of prey species in fecal samples (15 species at >1%) than in prey remains (5 species), suggesting that prey remains collected following a predation event may underestimate the contribution of some species in the diet. This finding could be due to several factors. First, each fecal sample likely represents multiple predation events due to the mixing that likely occurs during the several hours required for prey items to move through the digestive tract of a medium-sized cetacean [57]. The number of predation events represented in the fecal sample analysis is therefore larger than the number of fecal samples itself and this predation event homogenate may capture more diversity. In addition to incorporating more predation events, because the whales can travel relatively rapidly within their range, the fecal samples also integrate information over a longer time period and potentially greater spatial area than samples of prey remains. This could be an explanation for the higher proportion of

Chinook salmon in Puget Sound fecal samples than in Puget Sound prey remains. From October-December the whales generally make sporadic forays into Puget Sound from adjacent areas of their range [58]. These visits are generally brief (usually only 1–2 days) such that fall/early winter fecal samples collected in Puget Sound likely represented a combination of prey obtained in previous predation events in the Strait of Juan de Fuca or adjacent waters, rather than Puget Sound itself. In contrast, prey remains samples only represent diet specific to the area in which the samples were obtained. If the proportions of different prey species differ over space or time, this could at least partially explain how diet compositions estimated from prey remains differed from fecal samples.

Second, differences in estimated species composition between the two sample types could be due to biases associated with each sampling method, and the behavior of both the killer whales and prey species and the prey's distribution. For example, sampling of prey remains is likely to only give access to (and therefore enable identification of) prey species that are consumed at or near the surface, and are large enough to require tearing into pieces and/or sharing. In addition, some prey species or ages may be more suitable for prey sharing near the surface. Differences in prey morphology, such as how readily a prey species sheds scales (e.g., salmon compared to halibut or lingcod), may also account for some of the diet differences observed between prey remains and feces [8].

## Population origins of Chinook salmon prey

We found that SRKWs consumed a wide variety of Chinook salmon stocks during the non-summer seasons, and these stocks originated from a vast area of the Pacific Coast from nearly every major river system from the Sacramento River in California to the Taku River in northern British Columbia/SE Alaska, with Columbia River stocks being particularly prevalent. This diversity in the number of stocks consumed (14) contrasted markedly with the whales' Salish Sea (7) summer diet, which was primarily comprised of stocks from the Fraser River and Puget Sound [11].

SRKWs generally consumed Chinook salmon in relatively close proximity to the fish's natal rivers (e.g., Columbia River Chinook off the Washington coast, Puget Sound Chinook in Puget Sound), similar to them consuming mostly Fraser River Chinook salmon in summer in inland waters [11]. However, in non-summer months some Chinook salmon consumed were quite distant from their natal rivers, indicating that individuals from these stocks range widely, as has been seen with fishery recoveries of Coded Wire Tags (CWT) Chinook salmon [59]. Almost half of Chinook salmon prey remains samples collected during the fall in Puget Sound originated from river systems outside Puget Sound, from as far north as the Skeena River to as far south as the Columbia River. Chinook salmon consumed by J pod in the northern Georgia Strait in the winter included fish from a relatively close river system (lower Fraser River) as well as one that was relatively far (Taku River). However, overall the Chinook salmon stocks that J pod depends on in the winter remains unclear. Over 40% of the stocks consumed by K and L pods in Washington's outer coast waters in mid-winter/early spring originated from river systems very distant from the Washington coast, including as far south as California's Central Valley to as far north as the Taku River in northern British Columbia. The consumption of stocks that in many cases were distant from natal rivers illustrated that SRKWs rely on Chinook salmon stocks from a substantial portion of the range of Chinook salmon in western North America. The implication of a foraging strategy that relies on a portfolio of stocks whose upland or ocean distributions are located over a broad area is that this diversity may be advantageous by effectively dampening out some the annual fluctuations that naturally occur among these stocks.

The seasonal and spatial variation observed in Chinook salmon stock composition of the whales' outer coast prey was generally consistent with what is known about the ocean distribution and abundance patterns of Chinook salmon based on analyses of genetics [60] and CWT recoveries of fall Chinook salmon [61] in outer coast fisheries. Overall SRKWs consumed 14 Chinook salmon stocks on the Washington coast compared to 42 stocks collected in a troll fishery there [31, 60]. Columbia River Chinook salmon represented slightly over half (54%) of the prey remains samples we collected off the Washington coast, remarkably similar to the approximately 53.7% collected in outer coast troll fisheries based on genetic analysis [60], though higher compared to 30% based on CWT recoveries for fall Chinook salmon [61].

The relatively large proportion of Columbia River Chinook salmon stocks consumed by SRKWs was likely a function of three factors: the relatively large amount of time the whales spend near the Columbia River, the seasonal increase in fish aggregations associated with spawning, and the relatively large number of Chinook salmon returning to the Columbia River system. K/L pods consistently spent the majority of their time off the Washington coast during the times we collected samples, which is in concordance with acoustic recorder occurrence information [62] and satellite tag location data [63]. Secondly, the whales were consuming fish in the outer coast waters adjacent to the Columbia River during a season when some stocks (spring-run) would be expected to start moving into this area [64, 65]. Finally, the Columbia River system, despite its overall production being a fraction of its historic levels, still produces more Chinook salmon than any other system on the west coast of North America (see Fig 2 in [17]). Consequently, all these factors are expected to increase the whales' likelihood of encountering Chinook salmon from this system. This situation is similar to what we observed for this population in their summer range in inland waters where Chinook salmon stocks they consumed were generally in close proximity to their natal river, the Fraser River [11], which produces more Chinook salmon than Puget Sound or any of the other inland rivers in the Salish Sea.

## Implications of potential competition with Northern Resident killer whales

SRKWs and northern resident killer whales have been shown to be salmon specialists, with a strong selectivity for Chinook salmon [8–11]. These two adjacent, partially overlapping populations [66] potentially consume many of the same Chinook salmon stocks as most of these stocks migrate north following ocean emergence [59] before returning south to their natal rivers as adults. Limited diet data are available for NRKWs apart from summer data [67] that indicated this population to be consuming a similar number of stocks in summer [68] as SRKWs consumed in winter (this study, see Supplemental tables). A further comparison suggests that SRKWs have fewer potential stocks available to them when they are in the inland water portion of their summer range compared to NRKWs. This is likely the case in that most of the Chinook stocks consumed by SRKWs during summer come primarily from one basin, the Fraser River, that is only comprised of five stock groups [64], each with different temporal run timing [64], which was reflected in the observed peaks in consumption. The recent reduction in SRKW residency time in the Salish Sea [58] may be related to the recent general decline in most Chinook salmon stocks in the Fraser River [69]. Our data also provides some direct evidence of dietary overlap with NRKWs, suggesting possible competition between these two killer whale communities, as proposed in a previous study [17]. The combination of NRKWs removing Chinook salmon from stocks prior to, or in the same general time, and areas, as SRKWs potentially puts the latter population at a caloric disadvantage.

Fish age is highly correlated with size in salmon [70] and is therefore of importance for determining its caloric value. While there was dietary overlap in species (Chinook and chum salmon), and Chinook salmon stocks, consumed by SRKWs and NRKWs [8], a major

difference between our study and [8] was that our chum and Chinook salmon samples were from fish that were younger than those consumed by NRKWs (Fig 5A and 5B). Besides SRKWs generally consuming younger fish, of both Chinook and chum salmon, the youngest age class that they consumed was not present in the NRKW sample, and the oldest age fish in the NRKW samples were absent in SRKW samples. Potential reasons for the lack of older fish in SRKW diet could be due to prior consumption by competitors, e.g., NRKWs [71], or other predators of adult salmon [72]. Geographically, due to of most west coast originating Chinook salmon maturing in the waters of British Columbia and Alaska, prior to their southbound migration to their natal rivers [59], the generally more northerly occurring NRKWs have access to the older fish prior to SRKWs. NRKWs have been documented to select the not as plentiful, older, larger fish [8]. It is also possible that older age class fish do not occur in some of the stocks the SRKWs eat, although given the apparent overlap in stocks consumed by both populations this may be unlikely. Finally, Chinook salmon are getting smaller and younger over time [70] and the samples in [8] were collected between 2003 and 2005 whereas ours were collected later (2004 to 2017). The net result is that the consistent consumption of these smaller fish, which have a lower caloric value [73], may pose an additional challenge to the SRKW population's ability to meet their energetic needs.

## Conservation implications of Chinook salmon to Southern Resident killer whales

Our finding of year-round Chinook salmon consumption emphasizes the central importance of this prey species for SRKWs and suggests that conservation efforts to also increase

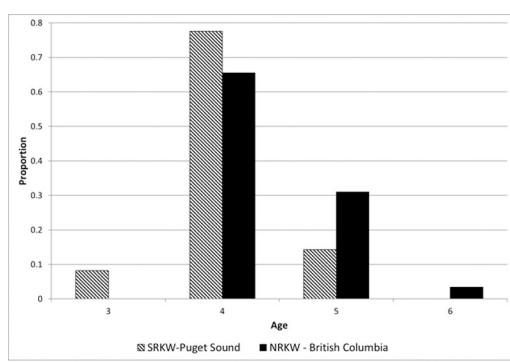

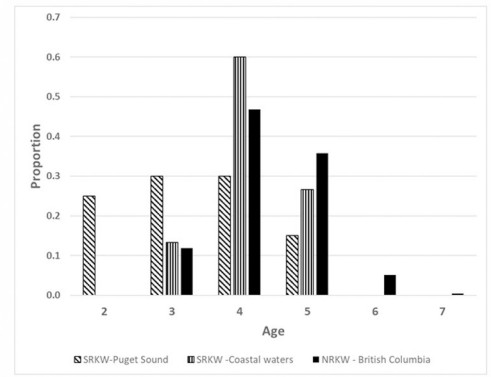

**Fig 5.** a. Age composition of chum salmon consumed by SRKW in the Puget Sound portion of our study area and primarily by NRKW in British Columbia [8]. b. Age composition of Chinook salmon consumed by SRKW in Puget Sound and the outer coast waters of our study and primarily by NRKW in British Columbia [8].

availability of Chinook salmon in the non-summer may be particularly important to this killer whale populations' recovery. However, the increased diet diversity in the winter months also underscores the importance of other species at particular times and in specific locations. Consequently, efforts to increase the availability of these other species, with appropriate temporal and spatial focus, should not be overlooked as an effective component of increasing prey for SRKWs.

Both K and L pods were documented to consume a diverse number of Chinook salmon stocks in outer coast waters, with stocks from the Sacramento River, Columbia River, and Puget Sound dominating their mid-winter/early spring diet. These basins all include Chinook salmon stocks that are listed as threatened or endangered under the ESA, and are themselves subject to extensive recovery efforts [16, 74]. Many Chinook salmon populations in these areas are now supplemented by extensive releases of fish from hatcheries, which can result in some relatively large spawning returns to these systems. Overall returns of Chinook salmon to the Columbia River were on average higher during the years of our study than other years since construction of the Bonneville Dam in 1938 [75], although abundance of many Chinook populations in the Columbia had declined well before 1938 [76]. In contrast, stocks from the Sacramento River had record low returns of Chinook during the years of our study, while returns to Puget Sound have been fairly stable [77]. These contrasting trends illustrate both the complexity of linking the whales' status directly to specific salmon stocks, and also the likely importance of providing a diverse portfolio of Chinook salmon stocks as prey. Differences in the observed patterns of distribution, prey consumption, and fecundity among the three SRKW pods also adds to the complexity of implementing effective management actions.

Prey limitations may be particularly acute for K and L pods which have exhibited lower fecundity compared to J pod [78]. J pod has a more northern winter range which overlaps with the growing northern whale population. Until recently, J pod had relatively high fecundity, accounting for six of the nine births between 2014 and 2015. The statistical association between prey and fecundity [23, 24] highlights the potential importance of prey availability as a risk factor, and our finding that Chinook salmon prominently appeared in the diet year-round suggests this relationship may be causal as has been suggested for Cook Inlet beluga whales [79]. However, while carrying capacity for the whales is likely driven by prey, linking Chinook salmon abundance to demographic rates is difficult for extremely small populations, when demographic stochasticity may be a large source of variation [80]. Although some of the documented reduced fecundity has been hypothesized to be associated with the need for specific stocks such as Interior Columbia River spring-run Chinook salmon [25], our results indicate that K and L pods forage on a broad array of species and Chinook salmon stocks. The differences observed between contaminant ratios in K/L versus J pod biopsy samples [81] demonstrates not only that population as whole uses an even wider array of prey and stocks but also that they have been doing so for quite some time. Our results are consistent with population modeling by [78] that suggested the best fit for SRKW fecundity and survivorship to be a model that included the largest suite of Chinook salmon stocks. Similar results have been observed with the Cook Inlet beluga population and their Chinook salmon prey [79].

## Opportunities and challenges for managing Chinook salmon to meet Southern Resident killer whale prey requirements

The contribution of Chinook salmon hatchery stocks in the whales' diet in fall through spring is likely far greater than in summer when the whales are in inland waters where their prey have been dominated by Fraser River stocks [11], a system comprised of stocks with minimal or no hatchery production for nearly all runs [69]. In contrast, most Chinook salmon stocks

consumed in outer coast waters in mid-winter/early spring were from the Sacramento and Columbia Rivers and Puget Sound, all of which are comprised of 50–80% hatchery fish [82]. Similarly, in Puget Sound Chinook, coho, and chum salmon stocks consumed by the whales in fall/early winter also have high proportions of hatchery fish [82]. Even in late summer when SRKW are in inland waters they prey on Lower Fraser Chinook salmon [11], the only Fraser River stock which has a relatively large hatchery component [69]. Overall, our results suggest that large scale hatchery programs for Chinook salmon that have been in place for decades [82] are a major source of prey for this population throughout a substantial portion of the year.

The large portion of hatchery Chinook in the mid-winter/early spring diet of at least K/L pods may allow for the potential to manage Chinook salmon stocks of particular importance to SRKWs to maximize their availability to this population. But such actions are not without risks. It is important to note that such an approach could be confounded by the role of burgeoning predator populations [17] and should consider potential risks to vulnerable salmon stocks from actions such as increases in hatchery production [83]. Further, some of these stocks have been identified as vulnerable to climate change [84]. Consequently, consideration should also be given to actively managing other prey species identified in this study to benefit SRKWs. As such, an ecosystem context is important to consider when directing recovery efforts to address limitations in the SRKWs' prey. Knowledge about these killer whales' fall, winter, and spring diets increases our understanding of complex food web interactions and can be applied to inform a conservation strategy to increase the prey available to this endangered killer whale population while considering roles and concerns for multiple species. Understanding which salmon species stocks can fill gaps in the SRKWs' prey base informs prioritization of salmon recovery actions (e.g., habitat restoration) and help evaluate impacts on killer whales and salmon from a variety of actions (e.g., fisheries, hatchery production). Data on diet across seasons and throughout their range will also inform habitat protection for the whales, including identification of biological features which are important to conservation and designation of Critical Habitat, (i.e., water quality, prey availability, and passage conditions) under the ESA [85].

## Conclusions

Although substantial new information has been gained on the diet of SRKWs in the fall, winter, and spring, data are still lacking for parts of the year and geographical ranges for some or all of the pods. Although summer diet in inland waters has been relatively well documented, in recent years SRKWs have been spending more time in the summer in outer coast waters, likely consuming a different, yet unknown suite of species and stocks. In addition, limited information is available on the fall diet of all three pods when they are outside of Puget Sound. Almost no information is available on the diet of J pod during most of the winter and spring, although the relatively high levels of PBDEs measured in blubber biopsies from this pod indicates that they forage on fish that remain within the Salish Sea for extended periods, likely including resident-type Chinook salmon [86]. The bulk of our outer coast samples from K/L pods were collected in 2013 and 2015, years that had relatively high returns of Chinook salmon to the Columbia River [75] such that the samples we collected may represent an unusual situation for the SRKWs' dietary assemblage in this part of their range. Since 2015, Chinook salmon returns coast-wide have been lower, due to the recent California drought, and warm ocean conditions [87], likely resulting in an adverse impact on juvenile salmon survival and thus future adult returns. While the SRKWs may have the option to increase consumption of non-salmonid prey (e.g., halibut, lingcod) that may not, or be less affected, by these environmental conditions; another potential response would be that they spend more time in other areas of their

range, although this assumes sufficient prey are available in other portions of their range. This sort of change in habitat use appeared to have occurred in 2007, a year of historically low salmon returns to the Columbia River that coincided with very limited acoustic detections of SRKW presence/occurrence near the Columbia River mouth [62]. The segregation of the population into distinct winter ranges with K and L pods more southerly, and J pod more northerly, may be due to lower prey densities and further supports the notion that winter is a time when prey are more scarce.

Continued monitoring of SRKW diet and occurrence patterns is warranted in order to better understand the resiliency of this population as the whales have limited options to respond to declines in their preferred prey. In addition, as salmon management and recovery actions are implemented to increase the number of salmon available for this killer whale population [88], continued diet monitoring will be important in assessing the adequacy of these actions and informing an adaptive approach to recovery of SRKWs.

## Supporting information

**S1 Table. Southern Resident killer whale fecal sample collection information.** Sample number, whale pod identification from genotyping, and proportional contribution of prey species by region of the study area where samples were collected.
(DOCX)

**S2 Table. Fish voucher specimen names.** Genbank accession number and species name for the 403 fish species found on the U.S. west coast that were used in the analyses of prey in Southern Resident killer whale fecal samples.
(DOCX)

**S3 Table. Southern Resident killer whale diet effort and sample summary.** Summary of sampling effort and sample type recovered from Southern resident killer whales during October to May 2004 to 2017 in Puget Sound (PS), and Juan de Fuca Strait/San Juan Islands (JdF/SJI), Northern Georgia Strait (NGS), and the outer coast waters of Washington, Oregon, and California (CW), study areas.
(DOCX)

**S4 Table. Observed and expected species composition of experimental controls for Southern Resident killer whale prey species.**
(DOCX)

**S5 Table. Southern Resident killer whale Chinook prey monthly stock composition.** Estimated mean percentage (±SE) of Chinook salmon stock composition of Southern Resident killer whale prey from Puget Sound (PS) and outer coast waters (CW) based on remains from prey capture event samples for February, March, and April. Chinook genetic stock groups are arranged from north to south. Chinook populations from [26], Appendix 1.
(DOCX)

## Acknowledgments

For efforts in the mid-winter/early spring in the outer coast waters and northern Georgia Strait we appreciate the all the dedicated efforts of the officers and crews of the NOAA vessels McArthur II and Bell M. Shimada. We are indebted to the Biowaves Inc. staff, in particular to the late Tom Norris who was instrumental in developing our ability to acoustically detect and track killer whales on the outer coast. We also appreciate the efforts of numerous field volunteers on these cruises. J. Foster, E. Falcone, and B. Rone assisted in small boat operations. In

inland waters we benefited from the whale presence alerts from Orca Network as well as fecal samples provided by S. Wasser and the numerous field volunteers. J. Samhouri provided constructive comments on the manuscript. E. Ward assisted with statistics. D. Holzer provided the map figure.

## Author Contributions

**Conceptualization:** M. Bradley Hanson, Candice K. Emmons, Linda K. Park, Jennifer Hempelmann.

**Data curation:** M. Bradley Hanson, Candice K. Emmons, Michael J. Ford, Meredith Everett, Kim Parsons, Jennifer Hempelmann, Donald M. Van Doornik, Gregory S. Schorr, Jeffrey K. Jacobsen, Mark F. Sears, Maya S. Sears, John G. Sneva.

**Formal analysis:** M. Bradley Hanson, Michael J. Ford, Meredith Everett, Kim Parsons, Jennifer Hempelmann, Donald M. Van Doornik, John G. Sneva.

**Funding acquisition:** M. Bradley Hanson.

**Investigation:** M. Bradley Hanson, Michael J. Ford, Meredith Everett, Kim Parsons, Linda K. Park, Jennifer Hempelmann, Donald M. Van Doornik, Gregory S. Schorr, Jeffrey K. Jacobsen, Mark F. Sears, Maya S. Sears, John G. Sneva, Robin W. Baird, Lynne Barre.

**Methodology:** M. Bradley Hanson, Michael J. Ford, Meredith Everett, Kim Parsons, Linda K. Park, Jennifer Hempelmann, Donald M. Van Doornik, John G. Sneva.

**Project administration:** M. Bradley Hanson, Linda K. Park.

**Resources:** M. Bradley Hanson.

**Supervision:** M. Bradley Hanson, Michael J. Ford, Linda K. Park.

**Validation:** M. Bradley Hanson, Michael J. Ford, Meredith Everett, Kim Parsons, Jennifer Hempelmann.

**Visualization:** M. Bradley Hanson, Michael J. Ford.

**Writing – original draft:** M. Bradley Hanson, Michael J. Ford, Robin W. Baird, Lynne Barre.

**Writing – review & editing:** M. Bradley Hanson, Candice K. Emmons, Michael J. Ford, Meredith Everett, Kim Parsons, Linda K. Park, Jennifer Hempelmann, Donald M. Van Doornik, Gregory S. Schorr, Jeffrey K. Jacobsen, Mark F. Sears, Maya S. Sears, John G. Sneva, Robin W. Baird, Lynne Barre.

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
