## [Decision Letter · Decision Letter 0]

26 Jun 2020

PONE-D-20-14399

Endangered predators and endangered prey: Seasonal diet of Southern Resident killer whales

PLOS ONE

Dear Dr. Hanson,

Thank you for submitting your manuscript to PLOS ONE. After careful consideration, we feel that it has merit but does not fully meet PLOS ONE’s publication criteria as it currently stands. Therefore, we invite you to submit a revised version of the manuscript that addresses the points raised during the review process.

In addition to the comments and suggestions from the two reviewers, I would like you to focus on the following improvements: 

- Clarify Terminology:

Select and use one designation for ‘prey remains’ is needed for clarity.

Define seasons  Do they refer to calendar seasons or are they tied to the ecology of the whales or of their prey?

- Provide more background Information:

The Northern Resident stock should be introduced somewhere in the Introduction or in the Methods (first mention is in the caption of Fig 8).

Finally, the authors refer to ‘whales’ instead of ‘killer whales’. I would suggest introducing the common name at the beginning of the paper, and using it consistently.

Readers unfamiliar with the SRKW and the NRKW could benefit from a brief introduction of their population trends, and the body condition and demography of these two ecotypes.  

- More effective use of tables:  The tables included in the ms are rather large and have many empty cells.  I would urge the authors to consider moving tables 1, 2, 3 and 4 to the supplementary materials section, and including more distilled tables (or figures) in the ms.  For instance, table 2 could be aggregated by season, rather than by month.

Additionally, in Table 2, the “+” sign in some of the cells is confusing, it would be better to provide a range of the individuals contributing to the fecal sample. 

- More effective use of figures:  The introductory map and the insert set the stage.  However, the other two maps (Figure 3 and 4) are not very effective because the points are small and overlay each other, and large scale maps are used to capture distributions with outliers.  I would suggest using a larger study are map (figure 1), where you define the various geographic zones, and use inserts to identify the geographic locations mentioned in the text. Then, given the detailed depiction os these geographic areas (and their inclusion in the summary tables), I would remove the two other maps.      

Figure - If you removed figure 3 and 4, you could addition a map showing the range of the two populations and the locations of some of the primary rivers.  Alternatively, if you wanted to highlight specific samples that were distinct / unique in the map, I would suggest you limit the mapping of the samples to few numbered and color-coded symbols.

Figure 8b – The y axis label should read “Proportion”. Please fix this typo.

We look forward to receiving your revised manuscript.

Kind regards,

David Hyrenbach, Ph.D.

Academic Editor

PLOS ONE

Journal Requirements:

2. We note that you are reporting an analysis of a microarray, next-generation sequencing, or deep sequencing data set. PLOS requires that authors comply with field-specific standards for preparation, recording, and deposition of data in repositories appropriate to their field. Please upload these data to a stable, public repository (such as ArrayExpress, Gene Expression Omnibus (GEO), DNA Data Bank of Japan (DDBJ), NCBI GenBank, NCBI Sequence Read Archive, or EMBL Nucleotide Sequence Database (ENA)). In your revised cover letter, please provide the relevant accession numbers that may be used to access these data. For a full list of recommended repositories, see http://journals.plos.org/plosone/s/data-availability#loc-omics or http://journals.plos.org/plosone/s/data-availability#loc-sequencing.

'The authors have declared that no competing interests exist.'

We note that one or more of the authors are employed by a commercial company: Bio-Waves, Incorporated.

Additional Editor Comments (if provided):

Reviewers' comments:

Reviewer's Responses to Questions

**Comments to the Author**

1. Is the manuscript technically sound, and do the data support the conclusions?

Reviewer #1: Yes

Reviewer #2: Yes

2. Has the statistical analysis been performed appropriately and rigorously? 

Reviewer #1: Yes

Reviewer #2: Yes

3. Have the authors made all data underlying the findings in their manuscript fully available?

Reviewer #1: Yes

Reviewer #2: Yes

4. Is the manuscript presented in an intelligible fashion and written in standard English?

Reviewer #1: Yes

Reviewer #2: Yes

5. Review Comments to the Author

Reviewer #1: Comments to the Authors

#General comments

The authors present the results of dietary analyses conducted for the (endangered) Southern Resident killer whale (SRKW) population in the eastern North Pacific Ocean. Using visual and molecular methods, they identify consumed prey species (salmonids and other fish) from prey remains and fecal samples collected from 2004 through 2007 in various areas and times of the year. The authors also aged and identified stocks of origin of the Chinook salmon prey consumed in order to assess the stocks of particular importance for the recovery of this killer whale population.

The manuscript is well structured with the information clear and presented in a straightforward manner. The analyses seem appropriate and the discussion is adequate, and not overly speculative. Overall, this study constitutes a good contribution to the killer whale literature and is of particular relevance and importance for the management and recovery of the SRKWs.

I recommend publication of the manuscript after some corrections. There is a need for a thorough check of all numbers throughout the ms (see detailed comments below). I also think that chosing one designation for ‘prey remains’ and sticking to it is needed for clarity. Seasons i.e. fall, winter, spring should be defined (whether they refer to typical calendar seasons or are tied to the whales’ or prey’s ecology). The Northern Residents should be introduced somewhere in the Introduction or in the Methods (first mention is in the caption of Fig 8). I am wondering if reference to Tables and Figures should be added in the Discussion. There is inconsistency with capital letters e.g., Chinook salmon, coho, chum that needs to be fixed throughout the manuscript. Also, too often the authors refer to ‘whales’ instead of ‘killer whales’. This may be confusing for people who may not be familiar with the killer whale literature.

Specific comments

Abstract

L29: What is ‘the Pacific coast’ – I suggest being more specific here; same with ‘coastal waters’ L31 – This would emphasize better the relatively large area investigated in this study

L30: Shouldn’t it be ‘fecals’ or ‘feces’ here?

L31: Add ‘of 2004-2017’ after May

L37: Add ‘salmon’ after ‘Chinook’

L38: Add ‘selectively’ before ‘consume’

L39: What is ‘coastal samples’?

L41: Could ‘effective’ be a better word than ‘useful’?

Introduction

L54-58: There may be missing info on the SRKW – Suggest adding a sentence or two telling about the status of the SRKW population (to highlight how critical – since L77 mentions ‘recovery’) and also emphasize the differences between the three different pods – could set the why it is important/relevant to look into dietary variations at the pod level in this study.

L65: Add ‘killer’ before ‘whale’

L67: ‘For characterizing these killer whales’ diet’

L69: Missing ref after ‘Salish Sea’, same for L71.

L70: Missing scientific name for coho salmon here

L77-80: I would also add to get insights into plasticity in feeding behavior (prey switch) and thus resiliency to declining preferred prey (further useful to model long-term population trends)

L81: Replace ‘the whales’ diet’ by ‘the SRKWs’ diet’

L83-84: Suggest rewording to ‘We further extended the analyses to genetically identify the stocks of origin for Chinook salmon prey’.

Methods

Line 73 (Caption Figure 1): add ‘between October and May from 2004 to 2017’.

Line 89: Replace ‘prey capture remains’ by ‘remains from prey capture events’.

L90: Add ‘between October and May’ before ‘from 2004 to 2017’

L93: An explanation of what a pod is and why it is relevant to investigate at the pod level would be useful, either here or in the introduction.

L93: ‘…, allowing for results to be presented for each pod, separately.

L102: replace ‘estimate stock composition of Chinook salmon prey samples’ by ‘determine stock identity of Chinook salmon prey samples’.

L103. End the sentence after ‘samples’, and start the new sentence with ‘Specifically, sample genotypes at 13 nuclear microsatellite DNA loci were compared to a coast-wide…’

L112: why pooled?

L143: coastal waters of?

Results

L154: ‘(JdF/SJI): 3 days and northern Georgia Strait (NGS): 3 days)’

L156: move ‘(36 days)’ after ‘California’

L157: More details would be useful here: samples from how many distinct prey capture events? 158? Were the 154 samples collected over the 156 days of killer whale encounters?

Same for the 81 fecal samples: from how many encounter days?

L160 (Table 1): What is the number after the year in column 1? Not clear. It would be helpful to mention in the table caption what a single prey sample refers to: does it represent a unique predation event? or are different types of remains e.g. when both scales and tissues were collected from a single prey capture event refered to as multiple samples? The total of prey samples comes to 155 whilst L157 indicates 154 samples collected. Needs correction.

L169: Adjust the percentage of scale/tissue samples after correction on total number of samples collected. As it stands here, it suggests 155 samples collected (and not 154).

L170: Revise accordingly here too.

L174 and 178: Add the number of samples collected before ‘scale/tissue samples’.

L181: Explain what are the pods somewhere in the Introduction or in the Methods.

L182: Again, there is contradiction between total count of prey samples collected in PS here (93) compared to Table 1 (94).

L188-190: Check correct numbers here also

L203: be consistent with having either ‘Table S3’ or ‘S3 Table’ (see L190)

L217-220: This is redundant with content from Figure 5 – There are a lot of Figures in the article. Maybe this one is not necessary, even though the visual representation may be better than having it in text.

L258-278: I suggest placing this section before the one on Species composition diet from fecal samples for consistency with the Method section.

L259: New contradiction with the number of prey samples collected

L262-264: Check correct numbers here, and also what months the authors refer to for the different seasons. Numbers seem to be incorrect.

L269: Replace ‘Table 1, Table 2’ by ‘Tables 1, 2’

L281: Table caption does not read well – suggest rewording

L292: Add ‘ were’ after 18.1% - Note that the total comes to 18.2% though

L335-337: there is no reference to NRKW before that

Discussion

L346: Replace ‘highlight that this species is a’ with ‘confirm this species as a’

L350: Remove one of the two ‘winter’

L351: Suggest replacing ‘this time of the year’ by ‘winter’ for clarity.

L356: add ref for summer

L360: be consistent with locations names – use either full names or abbreviations throughout the text

L370: wrong as written since ref #8 (Ford and Ellis 2006) is also about SRKWs – Suggest rewording

L371: Replace ‘whales’ by ‘killer whales’

L393: Add ‘that’ between ‘feces’ and ‘appeared’

L394: I suggest highlighting better that Halibut was reported as prey, not for any killer whale population in BC but specifically for resident type killer whales (Ford et al. 1998, Ford and Ellis 2006)

L398: Remove ‘been’ before ‘previously’

L422: add ‘itself’ after ‘samples’

L434: Remove ‘and’

L435: I suggest rewording to ‘…could be due to biases associated with sampling methods, and behavior of both killer whales and prey species.’

L437-41: heavy sentence. Needs rewording. It may be enough just to say that sampling of prey remains is likely to only give access to (and therefore enable identification of) prey species that are consumed at/near the surface, and are large enough to require tearing into pieces and/or sharing.

L450: ‘was primarily’ and ‘comprised of’

L471: I would replace ‘differences’ with ‘variations’

L471-474: Very long heavy sentence – Needs rewording.

L481: If Tables (or Figures) are going to be cited in Discussion, then they should be added many other places throughout the Discussion.

L499: Even though there appears to be strong selectivity for Chinook salmon, they do eat other salmonids and fish so I suggest down-grading this statement for more accuracy here. Also add ‘two’ after ‘These’

L501-504: Suggest rewording to: ‘Limited diet data are available for northern resident killer whales apart for summer data that indicated this population to be consuming a similar number…’

L504: ref (this study, Tables) after ‘consumed in winter’

L504-506: I don’t understand this. How is that?

L516: Add ‘is’ after ‘and’

L517: Reword to ‘for determining its caloric value?’

L520: add ‘from’ before ‘fish that were younger than..’

L521: Reword to: ‘Besides SRKWs generally consuming younger fish of both chinook and chum salmon, the youngest age class…’

L526: Reword to: ‘Geographically, due to most west coast originating Chinook salmon maturing in the waters of..’

L541: Add ‘also’ before ‘increase’ since it has been identified as a goal for the summer already (maybe add a ref fr that)

L542-545: this could be shortened since it largely repeats the previous sentence – I suggest ‘However, the increased dietary diversity in the winter months also underscores the importance of other species at particular times and in specific locations.’

L548: Reword to ‘Both K and L pods were documented…’

L562: Is reproduction relevant here?

L577: add ‘suggested’ before ‘the best fit’

L578: REPLACE ‘was’ by ‘to be’

L579: ‘…and their chinook salmon prey’

L604: Replace ‘in the whales’ prey’ with ‘in the SRKWs’ prey’ and ‘about the whales’ fall…’ with ‘about these killer whales’ fall…’, and ‘diet’ with ‘diets’

L606: Replace ‘the prey available to the whales’ with ‘the prey available to this critically endangered killer whale population’

L607-608: suggest remove since it is redundant with the following, more complete sentence

L609: Suggest replacing ‘fill gaps in the whales’ prey base’ with ‘fill gaps in the SRKWs’ prey base’

L611: Replace ‘whales’ with ‘killer whales’

L612: Replace ‘whales’ with ‘SRKWs’

L613: What are these biological features?

L616-618: Suggest rewording to: ‘Although substantial new information has been gained on the diet of the SRKWs in fall, winter and spring, data is still lacking for parts of the year and geographical ranges for some or all pods.

L619: Add ‘relatively’ before ‘documented’, Replace ‘the whales’ by ‘the SRKWs’

L628: Replace ‘for whales’’ with ‘for the SRKWs’’

L630: replace with ‘an adverse impact’?

L631: Replace ‘the whales’ by ‘the SRKWs’

L632: Replace ‘not necessarily’ by ‘that may not or less be’

L633: ‘these environmental conditions; another..’

L636: ‘detections of the SRKWs’ presence/occurrence near the’

L643: Replace ‘for this whale population’ with ‘for this killer whale population’

Reviewer #2: REVIEWER COMMENTS

Data presented in this paper represent a significant contribution to our understanding of SRKW diet and expand the available information for conservation and management decisions. The authors have collected a spatially diverse library of fecal samples and prey remains throughout the SRKW’s range and over a series of years to amass a detailed overview of SRKW diet and provide narrative on the composition of diet and potential implications for conservation. The methodologies are well described with attention to validation and control samples. The findings are informative with respect to the increased diversity in winter diet, differences between coastal and inland prey base, and the authors provide an interesting discussion on conservation implications of the data. The manuscript is well written, well referenced and a pleasure to read.

The comments on potential SRKW impacts from NRKW competition is intriguing and worthy of further discussion. Readers unfamiliar with the two populations of this fish-eating ecotype would benefit from knowing that the NRKW have maintained an increasing population trajectory of 2% annually and have a four-fold higher population (Towers 2018). In addition, information on body condition from both populations is available; these data are not discussed in this MS, but are relevant in relation to the interesting finding on inter-population differences in stock diversity, differences in age/size of prey and the implications to the energetic cost vs yield of foraging.

Improvement in the quality of the map and addition of the range of the two populations and the locations of some of the primary rivers (Columbia, Fraser, Thompson, Taku, Skeena, Snake, etc) would be beneficial.

The authors present interesting information on the relative proportion of wild vs hatchery stocks and the population’s dependence on these stocks in the spring and fall. In the summer months, prey are dominated by Fraser River stocks, a system comprised of stocks with minimal or no hatchery production. The condition of SRKW reputedly improves from their return to the waters of the Salish Sea in the early summer to their relative dispersal in the fall; these data indicate that this foraging period is primarily supported by wild salmon populations.

SPECIFIC LINE EDITS/COMMENTS

519 – data on NRKW age of prey from Ford and Ellis 2006 is from prey remains collected in 2003-2005. To better support the observation that NRKW consume older/larger Chinook, authors may consider application of a correction factor for the 15 year lag between the NRKW and SRKW data sets presented Fig 8a and B.

555 – suggest rewording for clarity – confusing statement.

624 – bioaccumulation of PBDEs provide insight into past foraging behaviours (Krahn et al, 2007) and these data represent foraging events that have occurred prior to 2007 (and with bioaccumulation and selective mobilization of lipids and thus contaminants, they may represent foraging events from decades in the past). Suggest modifying to indicate these data support the historic foraging conditions.

Figure 8b – y axis label should read “Proportion”

6. PLOS authors have the option to publish the peer review history of their article (what does this mean?). If published, this will include your full peer review and any attached files.

Reviewer #1: No

Reviewer #2: No

---

## [Author Response · Author response to Decision Letter 0]

27 Jan 2021

Below you will find my responses, in bold, to your comments and those of the reviewers for PONE-D-20-14399. In general I made nearly all of the requested changes. If you have any further questions please let me know.

Editor and reviewer comments

In addition to the comments and suggestions from the two reviewers, I would like you to focus on the following improvements: 

- Clarify Terminology:

Select and use one designation for ‘prey remains’ is needed for clarity. – I have changed all these designations to “prey remains”.

Define seasons Do they refer to calendar seasons or are they tied to the ecology of the whales or of their prey? - I have defined the seasons as “fall-early winter” (Oct-early January), “mid winter-early spring” (Late January – April) and “spring” (May) and further explained that these are tied to the ecology of the whales in that they reflect at least some part of their seasonal occurrence in some areas.

- Provide more background Information:

The Northern Resident stock should be introduced somewhere in the Introduction or in the Methods (first mention is in the caption of Fig 8). – A few sentences have been added to the introduction providing pertinent information about the northern resident killer whale population.

Finally, the authors refer to ‘whales’ instead of ‘killer whales’. I would suggest introducing the common name at the beginning of the paper, and using it consistently. – All “whales” have been changed to “killer whales”.

Readers unfamiliar with the SRKW and the NRKW could benefit from a brief introduction of their population trends, and the body condition and demography of these two ecotypes. - A section has been added to the introduction providing pertinent information about the northern resident killer whale population.

- More effective use of tables: The tables included in the ms are rather large and have many empty cells. I would urge the authors to consider moving tables 1, 2, 3 and 4 to the supplementary materials section, and including more distilled tables (or figures) in the ms. For instance, table 2 could be aggregated by season, rather than by month. – Per this request I reduced the number of tables in the manuscript to 3, moving table 1 to the supplemental materials and reducing tables 2 and 3 in size by consolidating data in the columns by season. 

Additionally, in Table 2, the “+” sign in some of the cells is confusing, it would be better to provide a range of the individuals contributing to the fecal sample. – I have changed all the “+” signs to “>” in order to more accurately convey that for some of the pooled samples, the prey species identified occurred in at least one of the pooled of samples but could have been present in more of them.

- More effective use of figures: The introductory map and the insert set the stage. However, the other two maps (Figure 3 and 4) are not very effective because the points are small and overlay each other, and large scale maps are used to capture distributions with outliers. I would suggest using a larger study are map (figure 1), where you define the various geographic zones, and use inserts to identify the geographic locations mentioned in the text. Then, given the detailed depiction os these geographic areas (and their inclusion in the summary tables), I would remove the two other maps. - Figure 1 was revised to incorporate the range of both southern and northern resident killer whales as well as better delineate areas referred to in the study area. 

Figure - If you removed figure 3 and 4, you could addition a map showing the range of the two populations and the locations of some of the primary rivers. Alternatively, if you wanted to highlight specific samples that were distinct / unique in the map, I would suggest you limit the mapping of the samples to few numbered and color-coded symbols. - Figures 3 and 4 were deleted and primary rivers added to the Figure 1

Figure 8b – The y axis label should read “Proportion”. Please fix this typo. - Corrected

5. Review Comments to the Author

Reviewer #1: Comments to the Authors

#General comments

The authors present the results of dietary analyses conducted for the (endangered) Southern Resident killer whale (SRKW) population in the eastern North Pacific Ocean. Using visual and molecular methods, they identify consumed prey species (salmonids and other fish) from prey remains and fecal samples collected from 2004 through 2007 in various areas and times of the year. The authors also aged and identified stocks of origin of the Chinook salmon prey consumed in order to assess the stocks of particular importance for the recovery of this killer whale population.

The manuscript is well structured with the information clear and presented in a straightforward manner. The analyses seem appropriate and the discussion is adequate, and not overly speculative. Overall, this study constitutes a good contribution to the killer whale literature and is of particular relevance and importance for the management and recovery of the SRKWs.

I recommend publication of the manuscript after some corrections. There is a need for a thorough check of all numbers throughout the ms (see detailed comments below). - All numbers have been checked and corrected

I also think that chosing one designation for ‘prey remains’ and sticking to it is needed for clarity. – all variations of references to prey remains have been changed to “prey remains”.

Seasons i.e. fall, winter, spring should be defined (whether they refer to typical calendar seasons or are tied to the whales’ or prey’s ecology). I have defined the seasons as “fall-early winter” (Oct-early January), “mid winter-early spring” (Late January – April) and “spring” (May) and further explained that these are tied to the ecology of the whales in that they reflect at least some part of their seasonal occurrence in some areas.

The Northern Residents should be introduced somewhere in the Introduction or in the Methods (first mention is in the caption of Fig 8). - A few sentences have been added to the introduction providing pertinent information about the northern resident killer whale population.

I am wondering if reference to Tables and Figures should be added in the Discussion. – I only referenced Fig 4 in the discussion as it pertains to a comparison of previously published data on ages of northern resident killer whale prey

There is inconsistency with capital letters e.g., Chinook salmon, coho, chum that needs to be fixed throughout the manuscript. – these have been corrected

Also, too often the authors refer to ‘whales’ instead of ‘killer whales’. This may be confusing for people who may not be familiar with the killer whale literature. - All “whales” have been changed to “killer whales”.

Specific comments

Abstract

L29: What is ‘the Pacific coast’ – I suggest being more specific here; same with ‘coastal waters’ - I have changed this L31 – This would emphasize better the relatively large area investigated in this study

L30: Shouldn’t it be ‘fecals’ or ‘feces’ here? – changed to feces

L31: Add ‘of 2004-2017’ after May - added

L37: Add ‘salmon’ after ‘Chinook’ added

L38: Add ‘selectively’ before ‘consume’ added

L39: What is ‘coastal samples’? – changed to “outer coast Chinook samples”

L41: Could ‘effective’ be a better word than ‘useful’? – changed as suggested

Introduction

L54-58: There may be missing info on the SRKW – Suggest adding a sentence or two telling about the status of the SRKW population (to highlight how critical – since L77 mentions ‘recovery’) and also emphasize the differences between the three different pods – could set the why it is important/relevant to look into dietary variations at the pod level in this study. - A few sentences have been added to the introduction providing pertinent background information about the southern resident killer whale population.

L65: Add ‘killer’ before ‘whale’ - added

L67: ‘For characterizing these killer whales’ diet’ – changed as suggested

L69: Missing ref after ‘Salish Sea’, same for L71. – added appropriate references to each sentence

L70: Missing scientific name for coho salmon here - added

L77-80: I would also add to get insights into plasticity in feeding behavior (prey switch) and thus resiliency to declining preferred prey (further useful to model long-term population trends) - added

L81: Replace ‘the whales’ diet’ by ‘the SRKWs’ diet’ – changed as suggested

L83-84: Suggest rewording to ‘We further extended the analyses to genetically identify the stocks of origin for Chinook salmon prey’. – changed as suggested

Methods

Line 73 (Caption Figure 1): add ‘between October and May from 2004 to 2017’. – changed as suggested

Line 89: Replace ‘prey capture remains’ by ‘remains from prey capture events’. – changed as suggested

L90: Add ‘between October and May’ before ‘from 2004 to 2017’ - added

L93: An explanation of what a pod is and why it is relevant to investigate at the pod level would be useful, either here or in the introduction. – The introduction was revised address this comment

L93: ‘…, allowing for results to be presented for each pod, separately. – changed as suggested

L102: replace ‘estimate stock composition of Chinook salmon prey samples’ by ‘determine stock identity of Chinook salmon prey samples’. – changed as suggested

L103. End the sentence after ‘samples’, and start the new sentence with ‘Specifically, sample genotypes at 13 nuclear microsatellite DNA loci were compared to a coast-wide…’ – changed as suggested

L112: why pooled? – added an explanation that was due to available resources at the time of analyses

L143: coastal waters of? – changed to outer coast waters of the U.S. west coast

Results

L154: ‘(JdF/SJI): 3 days and northern Georgia Strait (NGS): 3 days)’- changed

L156: move ‘(36 days)’ after ‘California’ - changed

L157: More details would be useful here: samples from how many distinct prey capture events? – each sample collected was from what appeared to be a single prey capture but as noted in the manuscript and detailed in S3 Table a second fish was sometimes identified

L158? Were the 154 samples collected over the 156 days of killer whale encounters?

Same for the 81 fecal samples: from how many encounter days? – we attempted to collect both prey remains and feces on all encounter days listed in S3 Table. Most days yielded one or both types of samples but breaking down by day would not be insightful due to the variability in conditions experienced on any given day. A variety of factors that potentially impacted the our ability to collect samples included, but was not limited to the length of the encounter or follow, weather conditions, behavior and speed of the animals, as well as the presence of vessels engaged in whale watching

L160 (Table 1): What is the number after the year in column 1? Not clear. – as shown in the column heading it is the year of the fall sampling with the winter/spring of the subsequent year after the hyphen

 It would be helpful to mention in the table caption what a single prey sample refers to: does it represent a unique predation event? or are different types of remains e.g. when both scales and tissues were collected from a single prey capture event referred to as multiple samples? - Sample type has been detailed in the table caption and column title was changed to reflect 

The total of prey samples comes to 155 whilst L157 indicates 154 samples collected. Needs correction. - Sample numbers have been revised

L169: Adjust the percentage of scale/tissue samples after correction on total number of samples collected. As it stands here, it suggests 155 samples collected (and not 154). - Sample numbers and percentages have been revised accordingly

L170: Revise accordingly here too. - Sample numbers and percentages have been revised accordingly

L174 and 178: Add the number of samples collected before ‘scale/tissue samples’. – these figures were deleted.

L181: Explain what are the pods somewhere in the Introduction or in the Methods. – this has been addressed in the introduction.

L182: Again, there is contradiction between total count of prey samples collected in PS here (93) compared to Table 1 (94). - Sample numbers have been revised

L188-190: Check correct numbers here also - Sample numbers have been revised

L203: be consistent with having either ‘Table S3’ or ‘S3 Table’ (see L190) – references to all supplemental tables have been revised to the correct format

L217-220: This is redundant with content from Figure 5 – There are a lot of Figures in the article. Maybe this one is not necessary, even though the visual representation may be better than having it in text. – Figure 5 has been deleted from the article

L258-278: I suggest placing this section before the one on Species composition diet from fecal samples for consistency with the Method section. – Earlier versions of the manuscript were configured as suggested but discussion by co-authors concluded that because the fecal samples played an important role in establishing the diversity of the winter diet that it should be featured up front in the discussion.

L259: New contradiction with the number of prey samples collected- Sample numbers have been revised

L262-264: Check correct numbers here, and also what months the authors refer to for the different seasons. Numbers seem to be incorrect.- Sample numbers have been revised

L269: Replace ‘Table 1, Table 2’ by ‘Tables 1, 2’ - not changed due to revisions of tables that were included

L281: Table caption does not read well – suggest rewording – caption reworded to clarify what table includes

L292: Add ‘ were’ after 18.1% - added - Note that the total comes to 18.2% though – 100 - 67.2 -14.7 = 18.1

L335-337: there is no reference to NRKW before that – northern residents are now described in the introduction

Discussion

L346: Replace ‘highlight that this species is a’ with ‘confirm this species as a’ – replaced as suggested

L350: Remove one of the two ‘winter’ – deleted as suggested

L351: Suggest replacing ‘this time of the year’ by ‘winter’ for clarity. - Replaced as suggested

L356: add ref for summer – was referenced in earlier part of the sentence

L360: be consistent with locations names – use either full names or abbreviations throughout the text – JdF/SJI was defined initially on line 179 pf the Results sections and have replaced other places this was spelled out

L370: wrong as written since ref #8 (Ford and Ellis 2006) is also about SRKWs – Suggest rewording – corrected this by adding #8 to portion referencing SRKW

L371: Replace ‘whales’ by ‘killer whales’ - replaced

L393: Add ‘that’ between ‘feces’ and ‘appeared’ – added ‘that’

L394: I suggest highlighting better that Halibut was reported as prey, not for any killer whale population in BC but specifically for resident type killer whales (Ford et al. 1998, Ford and Ellis 2006) – added reference to Ford and Ellis (2006) and revised Alaska reference as an unknown ecotype

L398: Remove ‘been’ before ‘previously’ – ‘been’ removed

L422: add ‘itself’ after ‘samples’ – added ‘itself’

L434: Remove ‘and’ – ‘and’ removed

L435: I suggest rewording to ‘…could be due to biases associated with sampling methods, and behavior of both killer whales and prey species.’ – reworded as suggested

L437-41: heavy sentence. Needs rewording. It may be enough just to say that sampling of prey remains is likely to only give access to (and therefore enable identification of) prey species that are consumed at/near the surface, and are large enough to require tearing into pieces and/or sharing. - reworded as suggested

L450: ‘was primarily’ and ‘comprised of’ – suggested changes made

L471: I would replace ‘differences’ with ‘variations’ – ‘differences’ replaced with ‘variations’

L471-474: Very long heavy sentence – Needs rewording. – was missing a couple of words which impacted clarity, have reworded to improve clarity 

L481: If Tables (or Figures) are going to be cited in Discussion, then they should be added many other places throughout the Discussion. – reference to Table 3 was deleted

L499: Even though there appears to be strong selectivity for Chinook salmon, they do eat other salmonids and fish so I suggest down-grading this statement for more accuracy here. – revised to reflect this sentiment Also add ‘two’ after ‘These’ – ‘two’ added

L501-504: Suggest rewording to: ‘Limited diet data are available for northern resident killer whales apart for summer data that indicated this population to be consuming a similar number…’ – reworded as suggested

L504: ref (this study, Tables) after ‘consumed in winter’ – cited as suggested

L504-506: I don’t understand this. How is that? – revised as follows: This A further comparison suggests that SRKWs have relatively fewer potential stocks available to them when they are in the inland water portion of their summer range compared to northern resident killer whalesNRKW. This is likely the case in that most of the Chinook stocks consumed by SRKWs during summer come mostly primarily from one basin, the Fraser River, which is only comprise of five stock groups [64] ,each with different temporal run timing [64] , which was reflected in the observed peaks in consumption [11] that are likely related to peak run timing for these few stocks [64].

L516: Add ‘is’ after ‘and’ – ‘is’ added

L517: Reword to ‘for determining its caloric value?’ – reworded as suggested

L520: add ‘from’ before ‘fish that were younger than..’- ‘from’ added as suggested

L521: Reword to: ‘Besides SRKWs generally consuming younger fish of both chinook and chum salmon, the youngest age class…’ - reworded as suggested

L526: Reword to: ‘Geographically, due to most west coast originating Chinook salmon maturing in the waters of..’- reworded as suggested

L541: Add ‘also’ before ‘increase’ since it has been identified as a goal for the summer already (maybe add a ref fr that) – ‘also’ added before ’increase’

L542-545: this could be shortened since it largely repeats the previous sentence – I suggest ‘However, the increased dietary diversity in the winter months also underscores the importance of other species at particular times and in specific locations.’ - reworded as suggested

L548: Reword to ‘Both K and L pods were documented…’ - reworded as suggested

L562: Is reproduction relevant here? – changed to fecundity

L577: add ‘suggested’ before ‘the best fit’ – added ‘suggested’ before ‘best fit’

L578: REPLACE ‘was’ by ‘to be’ – reworded as suggested

L579: ‘…and their chinook salmon prey’ - added words as suggested

L604: Replace ‘in the whales’ prey’ with ‘in the SRKWs’ prey’ - reworded as suggested, and ‘about the whales’ fall…’ with ‘about these killer whales’ fall…’, - reworded as suggested, and ‘diet’ with ‘diets’- reworded as suggested

L606: Replace ‘the prey available to the whales’ with ‘the prey available to this critically endangered killer whale population’- reworded as suggested

L607-608: suggest remove since it is redundant with the following, more complete sentence – removed sentence as suggested

L609: Suggest replacing ‘fill gaps in the whales’ prey base’ with ‘fill gaps in the SRKWs’ prey base’ - reworded as suggested

L611: Replace ‘whales’ with ‘killer whales’ - reworded as suggested

L612: Replace ‘whales’ with ‘SRKWs’- reworded as suggested

L613: What are these biological features? – added (i.e., water quality, prey availability, and passage conditions) to this sentence as these are the biological features identified for SRKW as part of the Critical Habitat designation under the ESA.

L616-618: Suggest rewording to: ‘Although substantial new information has been gained on the diet of the SRKWs in fall, winter and spring, data is still lacking for parts of the year and geographical ranges for some or all pods. – reworded as suggested

L619: Add ‘relatively’ before ‘documented’, - added ‘relatively’ before ‘documented’, Replace ‘the whales’ by ‘the SRKWs’ – replaced as suggested

L628: Replace ‘for whales’’ with ‘for the SRKWs’’ - replaced as suggested

L630: replace with ‘an adverse impact’?- replaced as suggested

L631: Replace ‘the whales’ by ‘the SRKWs’ replaced as suggested

L632: Replace ‘not necessarily’ by ‘that may not or less be’ replaced as suggested

L633: ‘these environmental conditions; another..’ replaced as suggested

L636: ‘detections of the SRKWs’ presence/occurrence near the’ - changed as suggested

L643: Replace ‘for this whale population’ with ‘for this killer whale population’ replaced as suggested

Reviewer #2: REVIEWER COMMENTS

Data presented in this paper represent a significant contribution to our understanding of SRKW diet and expand the available information for conservation and management decisions. The authors have collected a spatially diverse library of fecal samples and prey remains throughout the SRKW’s range and over a series of years to amass a detailed overview of SRKW diet and provide narrative on the composition of diet and potential implications for conservation. The methodologies are well described with attention to validation and control samples. The findings are informative with respect to the increased diversity in winter diet, differences between coastal and inland prey base, and the authors provide an interesting discussion on conservation implications of the data. The manuscript is well written, well referenced and a pleasure to read.

The comments on potential SRKW impacts from NRKW competition is intriguing and worthy of further discussion. Readers unfamiliar with the two populations of this fish-eating ecotype would benefit from knowing that the NRKW have maintained an increasing population trajectory of 2% annually and have a four-fold higher population (Towers 2018). – information relating to NRKWs was included in the introduction

In addition, information on body condition from both populations is available; these data are not discussed in this MS, but are relevant in relation to the interesting finding on inter-population differences in stock diversity, differences in age/size of prey and the implications to the energetic cost vs yield of foraging. – while an interesting aspect of a comparison of these 2 populations, addressing it is beyond the scope of this paper

Improvement in the quality of the map and addition of the range of the two populations and the locations of some of the primary rivers (Columbia, Fraser, Thompson, Taku, Skeena, Snake, etc) would be beneficial. – Fig1 has been revised to address this comment

The authors present interesting information on the relative proportion of wild vs hatchery stocks and the population’s dependence on these stocks in the spring and fall. In the summer months, prey are dominated by Fraser River stocks, a system comprised of stocks with minimal or no hatchery production. The condition of SRKW reputedly improves from their return to the waters of the Salish Sea in the early summer to their relative dispersal in the fall; these data indicate that this foraging period is primarily supported by wild salmon populations. – there is no specific request to address these comments and as such no changes were made

SPECIFIC LINE EDITS/COMMENTS

519 – data on NRKW age of prey from Ford and Ellis 2006 is from prey remains collected in 2003-2005. To better support the observation that NRKW consume older/larger Chinook, authors may consider application of a correction factor for the 15 year lag between the NRKW and SRKW data sets presented Fig 8a and B. – we are not aware of a “correction factor” that could be used to normalize the Chinook age data between these two data sets

555 – suggest rewording for clarity – confusing statement. – reworded for clarity

624 – bioaccumulation of PBDEs provide insight into past foraging behaviours (Krahn et al, 2007) and these data represent foraging events that have occurred prior to 2007 (and with bioaccumulation and selective mobilization of lipids and thus contaminants, they may represent foraging events from decades in the past). Suggest modifying to indicate these data support the historic foraging conditions. – added a sentence to indicate that these data support that the whales we likely doing this historically

Figure 8b – y axis label should read “Proportion” – revised as suggested

---

## [Editor Report · Decision Letter 1]

1 Feb 2021

Endangered predators and endangered prey: Seasonal diet of Southern Resident killer whales

PONE-D-20-14399R1

Dear Dr. Hanson,

We’re pleased to inform you that, after addressing the reviewer comments and suggestions, your manuscript has been judged scientifically suitable for publication and will be formally accepted for publication once it meets all outstanding technical requirements.

Kind regards,

David Hyrenbach, Ph.D.

Academic Editor

PLOS ONE

Additional Editor Comments (optional):

Thank you for effectively addressing all the reviewer comments and suggestions. 

Your work augmenting the background and the supplementary materials, greatly improved the ms.
---

## [Editor Report · Acceptance letter]

8 Feb 2021

PONE-D-20-14399R1 

Endangered predators and endangered prey: seasonal diet of Southern Resident killer whales 

Dear Dr. Hanson:

I'm pleased to inform you that your manuscript has been deemed suitable for publication in PLOS ONE. Congratulations! Your manuscript is now with our production department. 

Kind regards, 

on behalf of

Dr. David Hyrenbach 

Academic Editor

PLOS ONE